# Potent neutralization of Marburg virus by a vaccine-elicited antibody

Amin Addetia[1], Lisa Perruzza[2], Kaitlin Sprouse[1,3], Young-Jun Park[1,3], Matthew McCallum[1], Cameron Stewart[1], Bianca Partini[2], Jack T. Brown[1], Alessia Donati[2], Katja Culap[2], Alessio Balmelli[2], Bhavna Chawla[4], Swagata Kar[4], Michal Gazi[5], Kendra Alfson[5], Yenny Goez-Gazi[5], Ricardo Carrion Jr[5], Davide Corti[2], Fabio Benigni[2✉] & David Veesler[1,3✉]

Marburg virus (MARV) is a filovirus that causes severe and often lethal haemorrhagic fever[1,2]. Despite the increasing frequency of MARV outbreaks, no vaccines or therapeutics are licensed for use in humans. Here we designed mutations that improve the expression, thermostability and immunogenicity of the prefusion MARV glycoprotein (GP) ectodomain trimer, which is the sole target of neutralizing antibodies and vaccines in development[3–8]. We discovered a fully human, pan-marburgvirus monoclonal antibody, MARV16, that broadly neutralizes all MARV isolates, Ravn virus and Dehong virus with 40–100-fold increased potency relative to previously described antibodies[9]. Moreover, MARV16 provided therapeutic protection in guinea pigs challenged with MARV. We determined a cryogenic electron microscopy structure of MARV16-bound MARV GP. The structure shows that MARV16 recognizes a prefusion-specific epitope spanning GP1 and GP2, which blocks receptor binding and prevents conformational changes required for viral entry. We further determined the architecture of the MARV GP glycan cap, which shields the receptor-binding site, and identified architectural similarities with distantly related filovirus GPs. MARV16 and previously identified antibodies directed against the receptor-binding site[9–11] simultaneously bound MARV GP. These antibody cocktails required multiple mutations to escape neutralization by both antibodies, a result that paves the way for the development of MARV therapeutics resistant to viral evolution. MARV GP stabilization along with the discovery of MARV16 advance prevention and treatment options for MARV disease.

MARV belongs to the Filoviridae family and causes Marburg virus disease (MVD), which is characterized by a haemorrhagic fever with a case fatality rate ranging from 24 to 88%[1,2,12]. In recent years, multiple African countries have experienced MARV outbreaks, including Ghana in 2022, Equatorial Guinea in 2023, Rwanda in 2024 and Tanzania in 2023 and 2025 (refs. 8,13–16). Although recent outbreaks were contained, larger MARV outbreaks are likely to occur, similar to the 2013–2016 outbreak of the related Ebola virus (EBOV) in West Africa, which led to more than 28,600 infections and 11,325 casualties[17,18]. The recurring and frequent spillovers of MARV underscores the necessity for licensed vaccines and therapeutics, which are currently not available. By contrast, multiple countermeasures have been developed and approved for Ebola virus, including vaccines such as Ervebo (rVSV-ZEBOV) and Zabdeno/Mvabea (Ad26.ZEBOV/MVA-BN-Filo), as well as monoclonal antibody therapeutics including mAb114 (Ebanga) and the REGN-EB3 cocktail (Inmazeb). All of these therapeutics provide proof of concept for the successful development of filovirus countermeasures[19–22].

Vaccines and monoclonal antibodies in development against MARV target the viral GP because GP-directed antibodies have been suggested to be the primary correlate of protection against MVD[3]. The MARV GP is a homotrimeric protein anchored in the viral membrane and is responsible for recognition of the host receptor, NPC1, and subsequent membrane fusion that leads to viral entry[23,24]. The MARV GP is proteolytically cleaved by furin during viral morphogenesis to produce the GP1 and GP2 subunits that remain covalently linked by a disulfide bond[10,11,25]. GP1 comprises three domains: the core, the glycan cap and the mucin-like domain. The GP1 core contains the receptor-binding site (RBS), which is shielded from neutralizing antibodies by the highly glycosylated glycan cap and mucin-like domain[10,23,26,27]. In contrast to the EBOV or Sudan virus (SUDV) GPs, an ordered glycan cap has not been visualized for MARV GP, which suggests that the RBS might be more exposed for MARV GP compared with the EBOV and SUDV GPs[10,11,27,28]. GP2 is also composed of three domains: the wing, the core and the transmembrane domain. The GP2 core is the fusion machinery that promotes fusion of the viral and host membranes[29–31]. The wing domain is unique to the *Marburgvirus* genus and wraps around the GP equator, which was proposed to limit recognition by neutralizing antibodies[11].

[1]Department of Biochemistry, University of Washington, Seattle, WA, USA. [2]Humabs Biomed, a subsidiary of Vir Biotechnology, Bellinzona, Switzerland. [3]Howard Hughes Medical Institute, Seattle, WA, USA. [4]BIOQUAL, Rockville, MD, USA. [5]Texas BioMedical Research Institute, San Antonio, TX, USA. ✉e-mail: fbenigni@vir.bio; dveesler@uw.edu

Several MARV investigational vaccines have advanced to phase I or II clinical trials, with one of them, cAd3-Marburg, being used during the MARV outbreak in Rwanda in 2024 (refs. 3–8). These vaccines either display MARV GP on a viral vector or encode for MARV GP. These vaccines may benefit from the identification and inclusion of mutations that increase GP expression and prefusion stability, similar to the stabilizing mutations that were incorporated into vaccines for SARS-CoV-2 and for respiratory syncytial virus[32,33]. Only one stabilizing mutation that promotes trimer formation of the MARV GP ectodomain has been identified so far[34]. By contrast, several stabilizing mutations have been identified for the EBOV and SUDV GPs, which suggests that further optimization of prefusion MARV GP may be possible[34,35].

Monoclonal antibodies that target multiple MARV GP domains have been isolated from patients who recovered from MVD and from GP-immunized animals[9,36–38]. However, only antibodies that target the RBS had detectable, albeit weak, neutralizing activity against MARV[9]. Several of these RBS-directed antibodies have shown protective efficacy in animal models, with one of them, MR191, being the precursor to the investigational therapeutic monoclonal antibody MBP01 (refs. 9,39,40). Given that the neutralization potency of RBS-directed antibodies, including MR191, can be reduced by single mutations in the highly variable glycan cap[9], an antibody cocktail may prove to be more resistant to viral evolution.

Here we set out to identify MARV GP-stabilizing mutations to develop an immunogen that will enable subsequent discovery of vaccine-elicited antibodies that potently neutralize MARV. We identified two mutations in the MARV GP2 heptad repeat 1-C (HR1$_C$) in the GP2 core domain that increase expression, thermostability and immunogenicity of the prefusion ectodomain trimer while retaining its native prefusion structure and antigenicity. Immunization of a humanized transgenic mouse with prefusion-stabilized MARV GP ectodomain trimer enabled isolation of a potent neutralizing antibody, designated MARV16, that targets a prefusion GP epitope spanning the GP1 and GP2 subunits. We further demonstrate that MARV16 potently neutralizes historical and contemporary MARV variants and the related Ravn virus (RAVV) and Dehong virus (DEHV). We show that MARV16 protects guinea pigs against MVD when administered after MARV exposure. Finally, we show that MARV16 can bind to MARV GP simultaneously with RBS-directed antibodies, thereby providing a path towards the development of a therapeutic antibody cocktail for MVD.

## Expression of the MARV GPΔMuc ectodomain

To produce a soluble, prefusion MARV GP ectodomain trimer, we first designed a MARV GP construct that lacks the mucin-like domain (residues 257–425) and the transmembrane domain (residues 638–681) (Fig. 1a). The mucin-like domain was omitted from our construct because antibodies that target this domain are unlikely to be neutralizing[41,42]. We also incorporated three previously described mutations, W439A, F445G and F447N, which increase cleavage of precursor GP by furin[10], and the H589I mutation, which promotes the formation of monodisperse trimers in the absence of an exogenous trimerization domain[34]. Co-expression of this construct, termed MARV GPΔMuc$_{WT}$, with furin produced monodisperse trimers (Fig. 1b), as visualized by negative-stain electron microscopy. Binding to MR191 (Fig. 1c) confirmed proper folding and antigenicity of our MARV GPΔMuc$_{WT}$ ectodomain, which we used for a subsequent antibody discovery campaign.

## Identification of stabilizing mutations in HR1$_C$

Similar to other class I fusion proteins, MARV GP undergoes large-scale structural rearrangements to mediate membrane fusion with the GP2 HR1$_C$ region (residues 577–583), rearranging from a loop before fusion to a helix after fusion[10,11,29,30]. Previous studies have identified

EBOV GP2 HR1$_C$ mutations that stabilize the prefusion conformation and improve GP expression yields with and without the addition of a trimerization domain[34,35]. However, porting the EBOV GP T578P stabilizing mutation to MARV GP did not produce the same enhancement in expression[34]. Following our previous success in stabilizing the Langya virus G and Epstein–Barr virus gB proteins[43,44], we used ProteinMPNN[45] to assist the identification of stabilizing MARV GP HR1$_C$ mutations using the previously determined RAVV GP structure[11] as an input model. Identified mutations were visually inspected, and those that seemed compatible with the prefusion conformation were incorporated into the MARV GPΔMuc ectodomain (Fig. 1a,d). Individual substitution of T582P and F583V improved expression yields by 1.6-fold and 1.7-fold, respectively, compared with our original MARV GPΔMuc$_{WT}$, and led to a 2.5-fold enhancement when combined together (Fig. 1e). All three designed MARV GPΔMuc ectodomain constructs eluted as monodisperse species and at a similar retention volume to MARV GPΔMuc$_{WT}$ when analysed by size-exclusion chromatography (Fig. 1f). Electron microscopy imaging of negatively stained samples confirmed the monodispersity of the constructs (Fig. 1b). Moreover, retention of MR191 binding (Fig. 1c) indicated that introduction of the HR1$_C$ mutations did not alter the conformational integrity or antigenicity of MARV GP. Both the T582P (mean melting temperature ($T_m$) ± s.d. of 69.7 ± 0.2 °C) and F583V ($T_m$ of 71.2 ± 0.1 °C) mutations improved the thermostability of MARV GPΔMuc, increasing the $T_m$ by 0.9 and 2.4 °C, respectively. The T582P/F583V double mutant (designated MARV GPΔMuc$_{PV}$) led to a 2.6 °C greater melting temperature than the original MARV GPΔMuc$_{WT}$ ($T_m$ of 71.4 ± 0.2 compared with 68.8 ± 0.3 °C), which indicated that the mutant exhibited improved stability of the prefusion state (Fig. 1g).

We next assessed the immunogenicity of the stabilized MARV GP ectodomains by immunizing BALB/c mice ($n = 10$ per group) with three doses spaced 4 weeks apart of 0.1, 1 or 10 μg of MARV GPΔMuc$_{WT}$ or MARV GPΔMuc$_{PV}$ ectodomain with Addavax as an adjuvant (Fig. 1h). Both the MARV GPΔMuc$_{WT}$ and MARV GPΔMuc$_{PV}$ ectodomains induced potent serum binding titres after two doses, with a geometric mean titre (GMT) ranging from $3.9 \times 10^3$ to $3.7 \times 10^4$ (Fig. 1i and Extended Data Fig. 1). After three doses, mice immunized with the MARV GPΔMuc$_{PV}$ ectodomain exhibited serum binding titres (GMTs for 0.1, 1 and 10 μg of $2.1 \times 10^4$, $1.2 \times 10^5$ and $2.2 \times 10^5$, respectively) 1.9–4.0-fold higher than those for mice immunized with the MARV GPΔMuc$_{WT}$ ectodomain (GMTs for 0.1, 1 and 10 μg of $1.1 \times 10^4$, $5.0 \times 10^4$ and $5.4 \times 10^4$, respectively). Most mice immunized with 1 or 10 μg MARV GPΔMuc$_{WT}$ or MARV GPΔMuc$_{PV}$ ectodomain had detectable serum-neutralizing antibody titres after three doses (Fig. 1j and Extended Data Fig. 1). The serum-neutralizing antibody titres were similar for mice immunized with the MARV GPΔMuc$_{WT}$ ectodomain (GMTs for 0.1, 1 and 10 μg of 21, 84 and 58, respectively) or the MARV GPΔMuc$_{PV}$ ectodomain (GMTs for 0.1, 1 and 10 μg of 28, 57 and 87, respectively). These data suggest that the MARV GPΔMuc$_{PV}$ ectodomain is more immunogenic than the MARV GPΔMuc$_{WT}$ ectodomain and a promising vaccine candidate, particularly as current MARV vaccines minimally elicit neutralizing antibodies[3,7,46]. Moreover, strategies such as multimerization of the stabilized GP on nanoparticles[47–49] or delivering the stabilized GP as an mRNA vaccine[42,50] will probably aid in inducing a more robust neutralizing antibody response.

## Discovery of a potent MARV-neutralizing antibody

All monoclonal antibodies identified so far that target MARV GP display no or weak neutralization potency (half-maximal inhibitory concentration (IC$_{50}$) of >1 μg ml⁻¹) against MARV GP pseudoviruses and even weaker, if any, activity (IC$_{50}$ > 100 μg ml⁻¹) against authentic MARV[9,36–38]. To identify potent MARV-neutralizing antibodies, we used Alloy ATX transgenic mice that have human immunoglobulin loci encoding for the heavy chain and either the lambda (ATX-GL) or the kappa (ATX-GK)

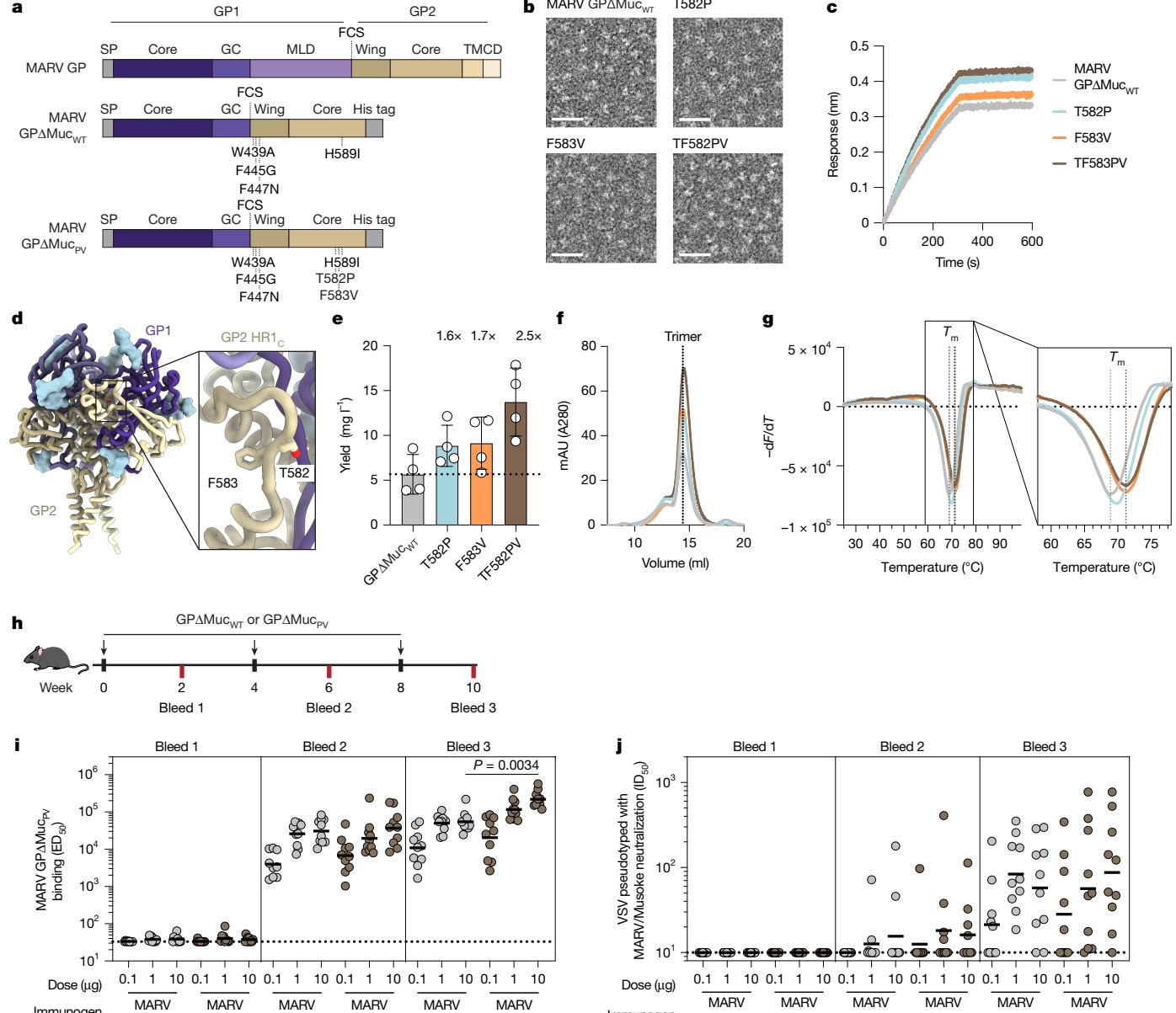

**Fig. 1 | Design of a prefusion-stabilized MARV GP. a**, Schematic of MARV GP domain organization and mutations included in MARV GPΔMuc ectodomain constructs. CD, cytoplasmic domain; FCS, furin cleavage site; GC, glycan cap; MLD, mucin-like domain; TM, transmembrane domain. SP, signal peptide. **b**, Negative-stain electron micrographs of MARV GPΔMuc constructs. Scale bar, 50 nm. Five micrographs were collected for each of the four biological replicates of each ectodomain. **c**, MARV GPΔMuc ectodomains binding to immobilized MR191 IgG assessed by BLI. **d**, Ribbon diagram of the RAVV GP (Protein Data Bank (PDB) identifier: 6BP2) highlighting the residues mutated in GP2 HR1$_c$. GP1, purple; GP2, gold; N-linked glycans, light blue surfaces. **e**–**g**, Recombinant production yields (**e**), size-exclusion chromatograms (**f**) and differential scanning fluorimetry analysis (**g**) of MARV GPΔMuc$_{WT}$ and MARV GPΔMuc mutants purified from Expi293 cells. The fold increase in yield relative to MARV GPΔMuc$_{WT}$ is displayed above the plot (**e**). The bar represents the mean yield ± s.d. across four biological replicates. mAU, milli-absorbance units. Data in **g** are shown as the first negative derivative of the fluorescence intensity (d*F*)

with respect to temperature (d*T*), and the $T_m$ of MARV GPΔMuc$_{WT}$ and MARV GPΔMuc$_{PV}$ are indicated with dotted vertical lines. Data presented in **c**, **f** and **g** are from one biological replicate and representative of three other biological replicates. Six technical replicates were conducted and averaged for each biological replicate in **g**. **h**, Scheme of the immunogenicity study design of MARV GPΔMuc ectodomains in BALB/c mice (*n* = 10 per group; bleed 2 serum could not be collected for 1 animal in the 0.1 µg MARV GPΔMuc$_{WT}$ group). **i**,**j**, MARV GPΔMuc$_{PV}$ serum binding titre (**i**) and neutralizing titre against VSV pseudotyped with the MARV/Musoke GP (**j**) respectively, presented as average ED$_{50}$ (half-maximal effective dilution) or ID$_{50}$ (half-maximal inhibitory dilution) values obtained from two biological replicates conducted in technical duplicate and using distinct batches of protein or pseudovirus. The limit of detection (ED$_{50}$ of 33 or ID$_{50}$ of 10) is indicated with a dotted line. Black lines indicate the geometric mean titre. Statistics were assessed using the Kruskal–Wallis test with Dunn's post-test comparing groups receiving identical doses of MARV GPΔMuc$_{WT}$ or MARV GPΔMuc$_{PV}$.

light chain. Two ATX-GL and two ATX-GK mice were immunized with MARV GPΔMuc$_{WT}$ for a total of three doses (Fig. 2a). Mice were killed 6 days after the last boost, and peripheral blood, spleen and lymph nodes were collected and cells were freshly isolated. MARV GPΔMuc$_{WT}$-specific memory B cells were selected by fluorescence-assisted cell sorting

(Extended Data Fig. 2), and variable domain (VH and VL) sequences were subsequently retrieved by PCR with reverse transcription PCR (RT–PCR).

We recovered ten antibodies that bound to MARV GPΔMuc$_{WT}$ (half-maximal effective concentration (EC$_{50}$) of 3.4–10.4 ng ml$^{-1}$)

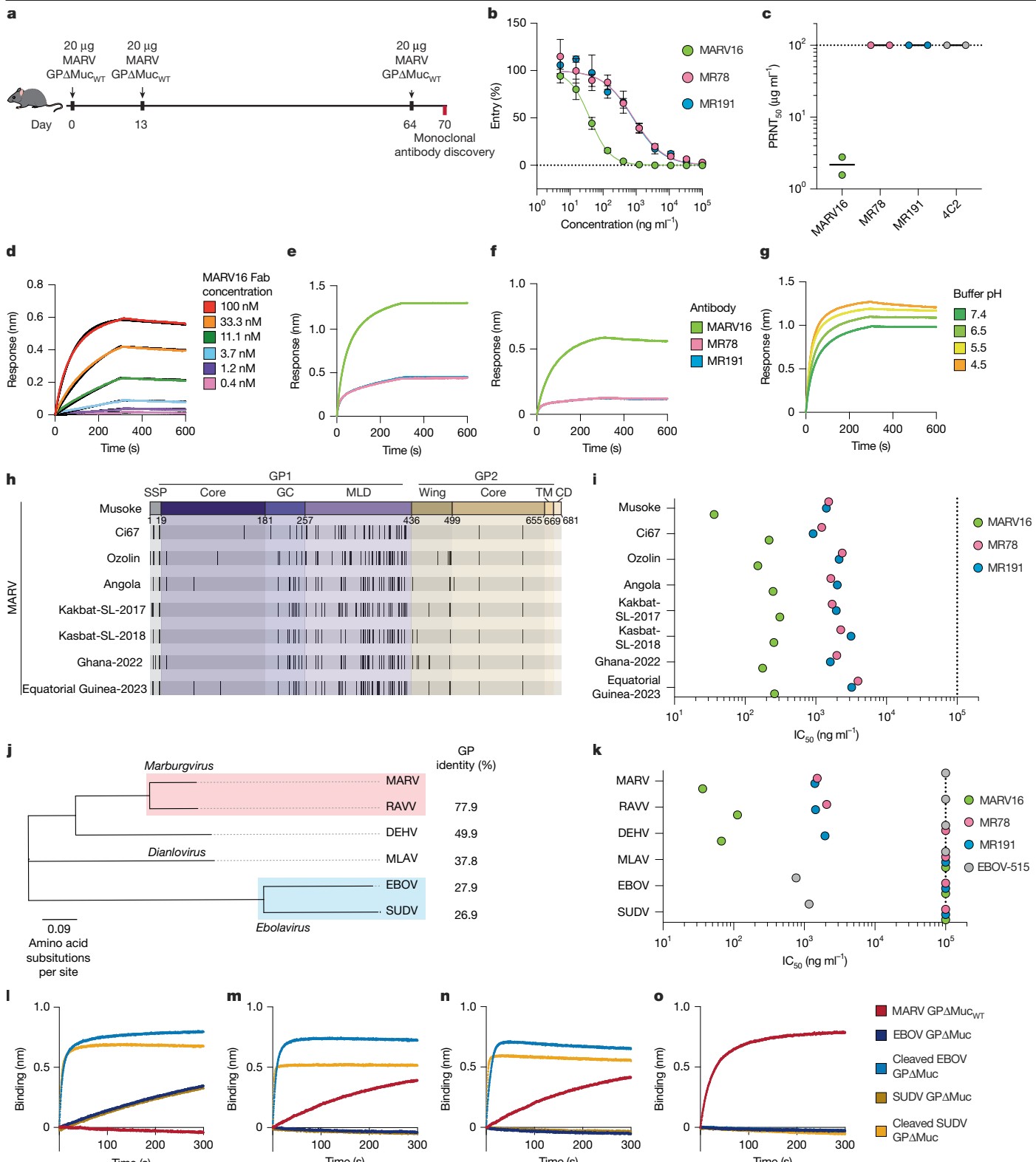

**Fig. 2 | See next page for caption.**

(Extended Data Table 1 and Extended Data Fig. 2), and one of them, designated MARV16, potently neutralized vesicular stomatitis virus (VSV) pseudotyped with the vaccine-matched MARV/Musoke GP ($IC_{50}$ of 36.4 ng ml$^{-1}$) (Extended Data Fig. 2). MARV16 is 42-fold and 39-fold more potent than the previously described RBS-directed neutralizing antibodies[9] MR78 ($IC_{50}$ of 1,520 ng ml$^{-1}$) and MR191 ($IC_{50}$ of 1,407 ng ml$^{-1}$), respectively, as measured side-by-side using MARV/Musoke GP VSV

pseudovirus (Fig. 2b). Furthermore, MARV16 neutralized authentic MARV/Musoke (50% plaque reduction neutralization test ($PRNT_{50}$) of 2.2 µg ml$^{-1}$), whereas no MR78-mediated or MR191-mediated neutralization was detected ($PRNT_{50}$ of >100 µg ml$^{-1}$ corresponding to the limit of detection of the assay) (Fig. 2c and Extended Data Fig. 2). These results establish MARV16 as a best-in-class MARV-neutralizing antibody.

**Fig. 2 | Discovery of a pan-marburgvirus antibody. a**, Schematic of the immunization schedule used to discover MARV GP-directed monoclonal antibodies (*n* = 4 mice). **b**, Dose–response neutralization curves for MARV16, MR78 and MR191 against VSV pseudotyped with the MARV/Musoke GP. Data are the mean ± s.e.m. from three technical replicates. Data are representative of 3–5 additional biological replicates. **c**, Neutralization potency of MARV16, MR78, MR191 and 4C2, a MERS-CoV antibody, against authentic MARV/Musoke. Data points reflect $PRNT_{50}$ values obtained from two biological replicates using distinct batches of IgGs. The black line indicates the mean $PRNT_{50}$ value. **d**, Binding affinity of the MARV16 Fab to immobilized MARV GPΔMuc$_{WT}$ measured using BLI. **e,f**, MARV16, MR78 and MR191 IgG (**e**) or Fab (**f**) binding to immobilized MARV GPΔMuc$_{WT}$ assessed by BLI. **g**, MARV16 IgG binding to immobilized MARV GPΔMuc$_{WT}$ at variable pH values using BLI. **h**, Schematic highlighting GP mutations (black vertical lines) in MARV variants relative to MARV/Musoke.

Residue numbers correspond to the MARV/Musoke GP. **i**, Neutralization potency of MARV16, MR78 and MR191 against VSV pseudotyped with the indicated MARV GP. **j**, Phylogenetic tree constructed using the amino acid sequences of related filovirus GPs with sequence identity relative to the MARV/Musoke GP shown to the right. **k**, Neutralization potency of MARV16, MR78, MR191 and EBOV-515 against VSV pseudotyped with the indicated filovirus GP. Data presented in **i** and **k** are averaged $IC_{50}$ values obtained from at least two biological replicates conducted in technical triplicate using distinct batches of IgG and pseudoviruses. **l–o**, MARV GPΔMuc$_{WT}$, EBOV GPΔMuc, SUDV GPΔMuc, thermolysin-cleaved EBOV GPΔMuc or thermolysin-cleaved SUDV GPΔMuc binding to immobilized EBOV-515 (**l**), MR191 (**m**), MR78 (**n**) or MARV16 (**o**) IgGs assessed with BLI. Data presented in **d–g** and **l–o** reflect one biological replicate and are representative of two biological replicates using distinct batches of proteins.

## MARV16 displays broadly neutralizing activity

We assessed the kinetics and affinity of binding of the MARV16 antigen-binding fragment (Fab) to immobilized MARV GPΔMuc$_{WT}$ by biolayer interferometry (BLI). The results revealed strong engagement characterized by single-digit nanomolar affinity (dissociation constant ($K_d$) of 1.35 nM) (Fig. 2d and Extended Data Table 2). Furthermore, MARV16 bound more strongly to MARV GPΔMuc$_{WT}$ than MR78 and MR191 in both IgG and Fab formats (Fig. 2e,f). Given that MARV enters target cells through fusion with the endosomal membrane induced by the low pH of late endosomes[51], we assessed the influence of pH on binding between MARV GPΔMuc$_{WT}$ and the three antibodies. MARV16 bound MARV GPΔMuc$_{WT}$ at comparable affinities at all four pHs tested (Fig. 2g). By contrast, MR78 and MR191 binding was unaltered at pH 7.4, 6.5 and 5.5, but was enhanced at pH 4.5 (Extended Data Fig. 3). We propose that these results may reflect increased accessibility of the RBS at lower pH values.

We next evaluated the neutralization breadth of MARV16 against seven historical and contemporary MARV isolates using VSV pseudotyped with the corresponding MARV GPs. These differ from that of the vaccine strain (MARV/Musoke) by 6.3–8.7% at the amino acid level (Fig. 2h), with most substitutions mapping to the glycan cap or mucin-like domain. MARV16 potently neutralized all 7 vaccine-mismatched MARV isolates, with $IC_{50}$ values ranging from 151 to 310 ng ml$^{-1}$ (Fig. 2i and Extended Data Fig. 4) and markedly outperformed MR78 and MR191 against all of these MARV isolates when assessed side by side.

Evaluation of MARV16-mediated neutralization breadth across the Filoviridae family revealed that MARV16 potently inhibited RAVV and DEHV VSV pseudoviruses, but not Měnglà virus (MLAV), EBOV or SUDV VSV pseudoviruses (Fig. 2j,k and Extended Data Fig. 4). Previous studies have shown that RBS-directed antibodies, including MR78 and MR191, recognize epitopes shared among all filovirus GPs but fail to neutralize *Ebolaviruses* owing to masking mediated by the glycan cap[9,10]. To determine whether MARV16 similarly recognizes a cryptic pan-filovirus epitope, we assessed whether MARV16 IgG binds the uncleaved and thermolysin-cleaved forms of the EBOV and SUDV GPΔMuc (that is, removing the glycan cap)[10] and compared it to EBOV-515, MR78 and MR191 using BLI. The *Ebolavirus* GP2-directed antibody EBOV-515 bound uncleaved and cleaved EBOV and SUDV GPΔMuc, but not MARV GPΔMuc$_{WT}$ (Fig. 2l). MR78 and MR191 bound MARV GPΔMuc$_{WT}$ and the cleaved EBOV and SUDV GPΔMuc, but not the uncleaved EBOV or SUDV GPΔMuc (Fig. 2m,n). MARV16 bound the MARV GPΔMuc$_{WT}$ but not EBOV or SUDV GPΔMuc, irrespective of their cleavage. This finding indicates that MARV16 does not cross-react with GPs from the *Ebolavirus* genus, which is most likely due to their extensive genetic divergence (Fig. 2j,o).

## Structural basis of MARV16-mediated neutralization

To understand the molecular basis of the potent neutralization of MARV by MARV16, we characterized the MARV GPΔMuc$_{WT}$ ectodomain bound to the MARV16 Fab using single-particle cryogenic electron microscopy (cryo-EM) and determined a structure at 2.6 Å resolution (Extended Data Table 3 and Extended Data Fig. 5). MARV16 recognizes an epitope that spans GP1 and GP2 (Fig. 3a,b and Extended Data Table 4). An average surface area of 1,100 Å$^2$ is buried at the interface between the epitope and the paratope with the majority of contacts with MARV GP contributed by the Fab heavy chain. GP2 accounts for about 75% of the epitope buried surface area and is recognized by all three complementarity-determining regions (CDRs) of the MARV16 heavy chain. CDRH3 residues recognize MARV GP2 through hydrogen bonds, salts bridges and van der Waals interactions, including CDRH3 R99 and D105 forming salt bridges with GP2 E515 and K550, respectively, and hydrogen bonding of CDRH3 N101 with the GP2 N551 side chain and of the CDRH3 W102 indol with GP2 N554 (Fig. 3c). CDRH1 and CDRH2 form extensive interactions with GP2, such as hydrogen-bonding between the CDRH2 S52 and S54 side chains and GP2 D513, CDRH2 Y57 and the backbone amide and carbonyl oxygen of GP2 R517, and the CDRH2 Y59 and GP2 R517 side chains. The CDRH1 T33 side chain interacts with the GP2 E515 side chain. The MARV16 light chain also interacts with GP2 primarily through CDRL3 involving D93 salt bridged to GP2 R517 and hydrogen bonding of the S91 and Y92 backbone carbonyls with the GP2 K550 side chain (Fig. 3d). MARV GP1 recognition is primarily mediated through MARV16 CDRH2 with the S54 and S56 backbone carbonyls hydrogen-bonded to the GP1 K90 and K120 side chains, respectively, and hydrogen bonding of the CDRH2 Y57 and GP1 E87 side chains (Fig. 3e). These extensive contacts explain the strong MARV16 binding affinity and the conservation of interface residues among MARV isolates, RAVV and DEHV explains the pan-marburgvirus neutralizing activity of this antibody (Fig. 2h–k and Extended Data Fig. 6). Multiple epitope residue substitutions explain the lack of EBOV and SUDV VSV neutralization mediated by MARV16 (Fig. 2j,k).

In the previously determined RAVV GP–MR191 structure, the GP2 wing partially obstructs the MARV16 epitope[11]. In our structure, the GP2 wing is disordered and the amino-terminal residues of the GP2 core shift by up to 18 Å relative to their position in the RAVV GP–MR191 structure, which enables binding of the MARV16 Fab (Extended Data Fig. 6). These data indicate that the wing and N terminus of the GP2 core are flexible and do not completely shield the GP from neutralizing antibodies. Compared with structures of previously characterized anti-EBOV GP antibodies, MARV16 shares a similar binding mode to the EBOV GP-directed neutralizing antibodies ADI-15946, EBOV-515 and EBOV-520 (Extended Data Fig. 6), which have been suggested to neutralize EBOV by tethering GP1 and GP2 in the prefusion conformation[52–55]. Our structural data suggest that MARV16 locks the MARV GP1 and GP2 interface through contacts with residues that are rearranged during fusogenic conformational changes that lead to membrane fusion[30]. Furthermore, comparison with the NPC1-bound EBOV GP structure[26,56] indicates that MARV16 would also interfere with receptor binding, as the heavy chain variable domain would sterically clash with the NPC1

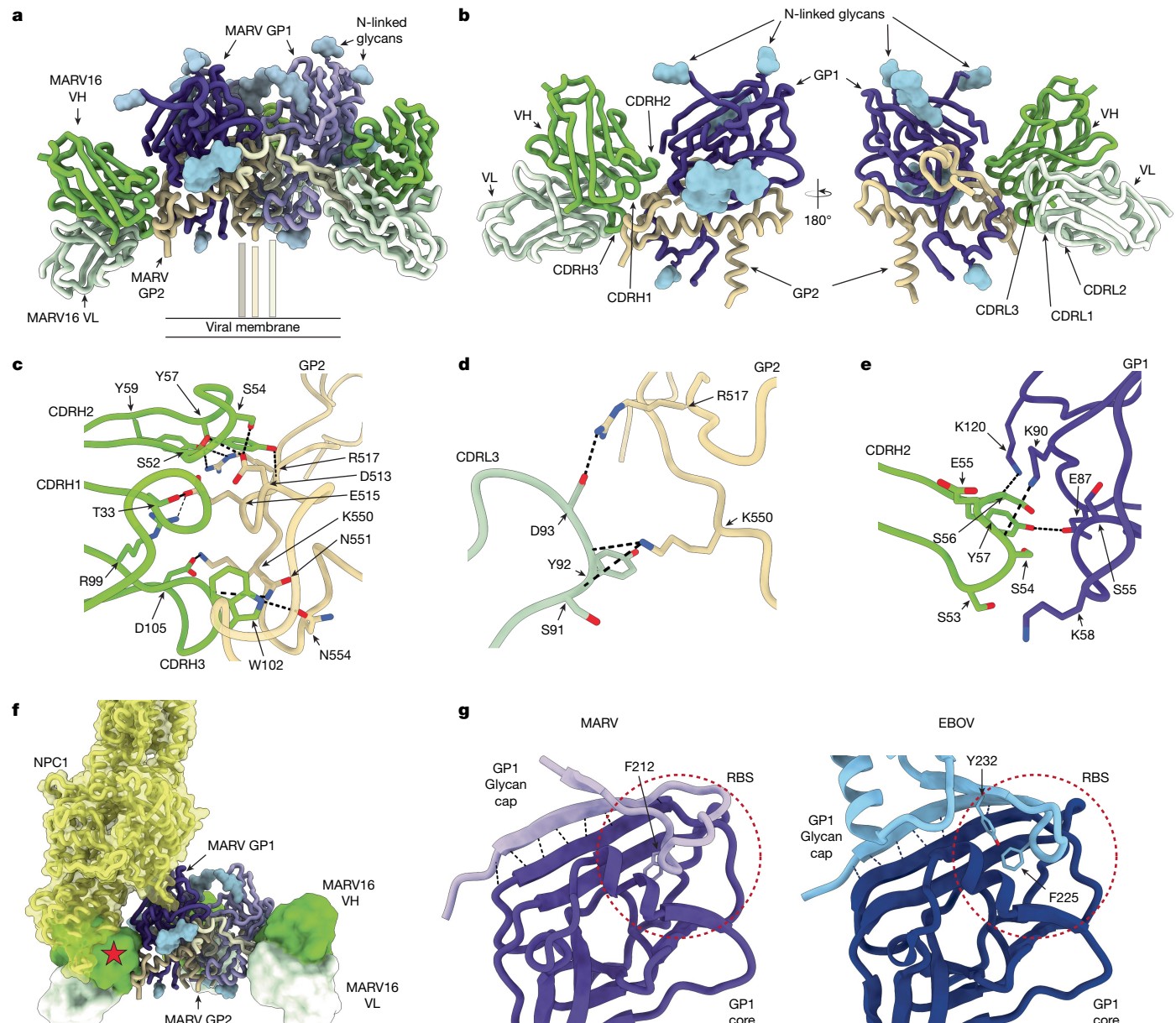

**Fig. 3 | Molecular basis of MARV16 neutralization. a**, Ribbon diagram of the cryo-EM structure of the MARV GPΔMuc$_{WT}$ ectodomain in complex with three MARV16 Fab fragments. Only the Fab variable domains were modelled into the density. MARV GP1 and GP2 are shown in shades of purple and gold, respectively, MARV16 VH is shown in green and MARV16 VL is shown in light green. N-linked glycans are rendered as light blue surfaces. **b**, Ribbon diagram of a single MARV GP protomer in complex with one MARV16 Fab. **c–e**, Zoomed-in views of interactions between MARV16 CDRH1–CDRH3 and MARV GP2 (**c**), MARV16 CDRL3 and MARV GP2 (**d**) and MARV16 CDRH2 and MARV GP1 (**e**). Selected hydrogen bonds and salt bridges are denoted with black dashed lines. **f**, Binding modes of the NPC1 receptor (yellow) and MARV16 (green/light green) to MARV GP. The position of NPC1 was determined by superimposing the EBOV GP–NPC1 (PBD: 5JNX) and MARV GP–MARV16 Fab structures. The EBOV GP trimer and the region of the MARV GP glycan cap resolved in our structure (residues 191–219) are not shown for clarity. The red star denotes steric clashes. **g**, Ribbon diagrams of MARV GP1 and EBOV GP1 (PDB: 3CSY). The MARV GP1 core and glycan cap are shown in dark and light purple, respectively. The EBOV GP1 core and glycan cap are shown in dark and light blue, respectively. Hydrogen bonds between the glycan cap and core are indicated with black dashed lines, and glycan cap aromatic residues inserted in the RBS are shown in stick representation and labelled. The position of the experimentally determined RBS for the EBOV GP and that of the putative RBS for the MARV GP are indicated with red dashed circles.

N-terminal domain (Fig. 3f), as is also the case for ADI-15946, EBOV-515 and EBOV-520.

## Resolving the MARV GP glycan cap

The discovery of neutralizing antibodies that target the MARV GP RBS from individuals infected with MARV suggests that the glycan cap might not be ordered and therefore does not shield the RBS[9,36]. Our cryo-EM map resolved density near the RBS that corresponded to residues 191–219 of the GP1 glycan cap, consistent with partial shielding of the RBS by the glycan cap in a way reminiscent of that observed for the EBOV GP (Extended Data Fig. 6). Indeed, the EBOV GP glycan cap interacts with the GP1 core through β-strand augmentation and insertion of F225 and Y232 into the RBS[27]. Our structure reveals that the MARV GP architecture is highly similar to that of the EBOV GP, sharing the β-strand augmentation and insertion of residue F212 into the MARV RBS (Fig. 3g). The possibly tighter anchoring of the EBOV GP glycan cap to the RBS, relative to MARV GP, may cause the latter

region to be more mobile and easily displaced than the EBOV glycan cap. This finding explains why MR78 and MR191 neutralize MARV, but not EBOV.

## Non-neutralizing antibodies bind the wing and HR2

We next mapped MARV GP epitopes recognized by the nine non-neutralizing monoclonal antibodies discovered through immunizing Alloy ATX mice. The antibodies clustered into four distinct binding groups on the basis of epitope binning performed using BLI (Extended Data Fig. 7). Group I consisted of MARV4, MARV12, MARV18, MARV21 and MARV23, whereas group II consisted of MARV11 and MARV14. MARV7 and MARV20 did not compete with the other antibodies analysed, which indicated that each of these two antibodies target distinct MARV GP antigenic sites. We then used electron microscopy of negatively stained samples to identify the MARV GP epitopes recognized by the four binding groups. Group I antibodies, represented by MARV18, binds HR2 (Extended Data Fig. 7). By contrast, group II antibodies, represented by MARV14, and MARV7 bind the GP2 wing (Extended Data Fig. 7). The epitope targeted by MARV20 could not be resolved, which may be due to MARV20 targeting a flexible region on the MARV GP (Extended Data Fig. 7).

## MARV16 protects against MVD in vivo

To assess the protective efficacy of MARV16, we challenged guinea pigs ($n = 6$ per group) with 1,000 plaque-forming units (PFU) of guinea pig-adapted MARV/Angola and administered 10 mg of MARV16 (human IgG1) 1, 2 or 4 days post-infection (d.p.i.) (Fig. 4a). An additional six guinea pigs were administered an isotype control monoclonal antibody 1 d.p.i. All six animals from the control group died by 13 d.p.i. and exhibited high plasma viral loads, weight loss (≥5% decrease) and increasing clinical scores after infection (Fig. 4b–f). Moreover, 3 out of these 6 animals displayed increased body temperatures (≥1.1 °C increase) after infection. For the MARV16-treated guinea pigs, the following percentage of animals survived: 33% (2 out of 6) of animals that received MARV16 1 d.p.i.; 83% (5 out of 6) of animals that received MARV16 2 d.p.i.; and 50% (3 out of 6) of animals that received MARV16 4 d.p.i. (Fig. 4b). We did not detect MARV16 plasma binding titre 1 day after treatment for two guinea pigs that received MARV16 1 d.p.i. (and died at 7 and 10 d.p.i.) and for one animal that received MARV16 4 d.p.i. (and died at 10 d.p.i.). These results suggest that in these animals, MARV16 was sequestered at the injection site and cleared or 'soaked up' immediately by the challenge virus before reaching the blood stream. After excluding these animals from the analysis, 50% (2 out of 4) of animals that received MARV16 1 d.p.i. and 60% (3 out of 5) of animals that received MARV16 4 d.p.i. survived the MARV challenge (Fig. 4b). Sixty percent of the surviving MARV16-treated guinea pigs experienced weight loss before recovering, whereas 40% did not experience any weight loss. Furthermore, 7 out of the 10 surviving guinea pigs had transient elevation of body temperatures, and all surviving animals exhibited low clinical illness scores (Fig. 4d–f). We observed 2-log, 3-log and 1-log reductions in MARV viral loads at 5 d.p.i. for animals treated with MARV16 at 1, 2 or 4 d.p.i., respectively (geometric mean viral loads of $1.2 \times 10^5$, $9.6 \times 10^3$ and $1.9 \times 10^6$ genome equivalents (GE) per ml, respectively) compared with guinea pigs that received the isotype control (geometric mean viral loads of $2.8 \times 10^7$ GE per ml) (Fig. 4c). These data indicate that MARV16 provides therapeutic protection against MARV challenge.

We next assessed the ability of MARV16 to trigger activation of FcγRIIa (H131) and FcγRIIIa (V158) as surrogate assays for antibody-dependent cellular phagocytosis (ADCP) and antibody-dependent cellular cytotoxicity (ADCC), respectively. MARV16 did not activate either FcγRIIa or FcγRIIIa (Extended Data Fig. 8). This result suggests that the observed protection induced by MARV16 originates solely from direct viral

neutralization. We propose that the introduction of modifications to the crystallizable fragment (Fc), as done for other MARV antibodies[37,39,40], to promote effector functions may further improve the therapeutic efficacy of MARV16.

## Formulation of MARV-neutralizing antibody cocktails

Antibody cocktails composed of two or more monoclonal antibodies that target nonoverlapping neutralizing epitopes are frequently used as antiviral therapeutics as they promote greater resistance to viral evolution than single monoclonal antibodies[57]. As MARV16 binds to a distinct epitope on MARV GP to that of the RBS-directed MR78 and MR191, we reasoned that MARV16 and MR78 or MR191 may simultaneously bind the MARV GP. To examine this possibility, we performed a competitive binding assay with MARV16 and MR78 or MR191 and observed that MARV GPΔMuc$_{WT}$ could bind MR78 or MR191 after binding MARV16 (Fig. 5a). Furthermore, three MARV16 Fab fragments and three MR78 or MR191 Fab fragments could simultaneously bind to the prefusion MARV GPΔMuc$_{WT}$ ectodomain trimer, as visualized by single-particle electron microscopy analysis of negatively stained samples (Fig. 5b,c).

To evaluate the barrier to escape from antibody-mediated neutralization, we first passaged replication-competent VSV encoding the MARV/Musoke GP instead of the VSV G protein in the presence of MARV16 alone. Two independent selection experiments were performed using two separately plaque-purified VSV-MARV/Musoke GP isolates, designated 2B2 and 2B4. The input 2B2 virus contained two nonsynonymous GP mutations mapping to the signal peptide (K2E) and the mucin-like domain (I381R), whereas the input 2B4 virus solely contained the I381R GP mutation. Both selection experiments using MARV16 alone produced a single escape mutation, A514T, in the GP2 core (Fig. 5d). These data are consistent with previous work on monoclonal antibodies, including MR78 and MR191, which demonstrated that a single mutation in the target viral antigen can enable it to escape single monoclonal antibodies[9,57].

To demonstrate that an antibody cocktail formulated with MARV16 and an RBS-directed antibody can prevent escape by a single MARV GP mutation, we next passaged the VSV-MARV/Musoke GP isolates in the presence of antibody cocktails composed of MARV16 and MR78 or MARV16 and MR191. In contrast to MARV16 alone, two or more mutations were required for the isolates to escape from the MARV16–MR78 and MARV16–MR191 antibody cocktails. Selection using the 2B2 isolate in presence of the MARV16–MR78 cocktail led to the identification of three escape mutations, Q128P, F447S and K550E, in the GP1 core, the mucin-like domain and the GP2 core, respectively (Fig. 5e). Using the 2B4 isolate for the MARV16–MR78 selection experiments produced two escape mutations, Y197C and A514E, in the glycan cap and GP2 core, respectively (Fig. 5e). For the MARV16–MR191 cocktail, three escape mutations were identified using the 2B2 isolate, which mapped to the glycan cap (C226R), the wing (L448P) and the GP2 core (K550E) (Fig. 5f). We identified two escape mutations, L451S and A514T, in the wing and the GP2 core, respectively, for the 2B4 isolate passaged in the presence of MARV16–MR191 (Fig. 5f). Our data indicate that mutations in the GP2 core probably affected recognition by CDRH1 and CDRH2 (A514T/E) or by CDRH3 and CDRL3 (K550E) (Fig. 5g) and led to MARV16 escape. Consistent with previous selection experiments[9], mutations in the RBS (Q128P or N129S in previous work) or mutations that probably prevented displacement of the glycan cap or wing (Y197C, C226R, L448P and L451S or S220P, C226Y and P455L in previous work[9]) led to MR78 and MR191 escape (Fig. 5g). These data indicate that MARV16 and an RBS-directed antibody can be used together in a therapeutic antibody cocktail that requires multiple GP mutations to escape neutralization by both antibodies, thereby creating a MARV therapeutic with increased resilience to viral evolution.

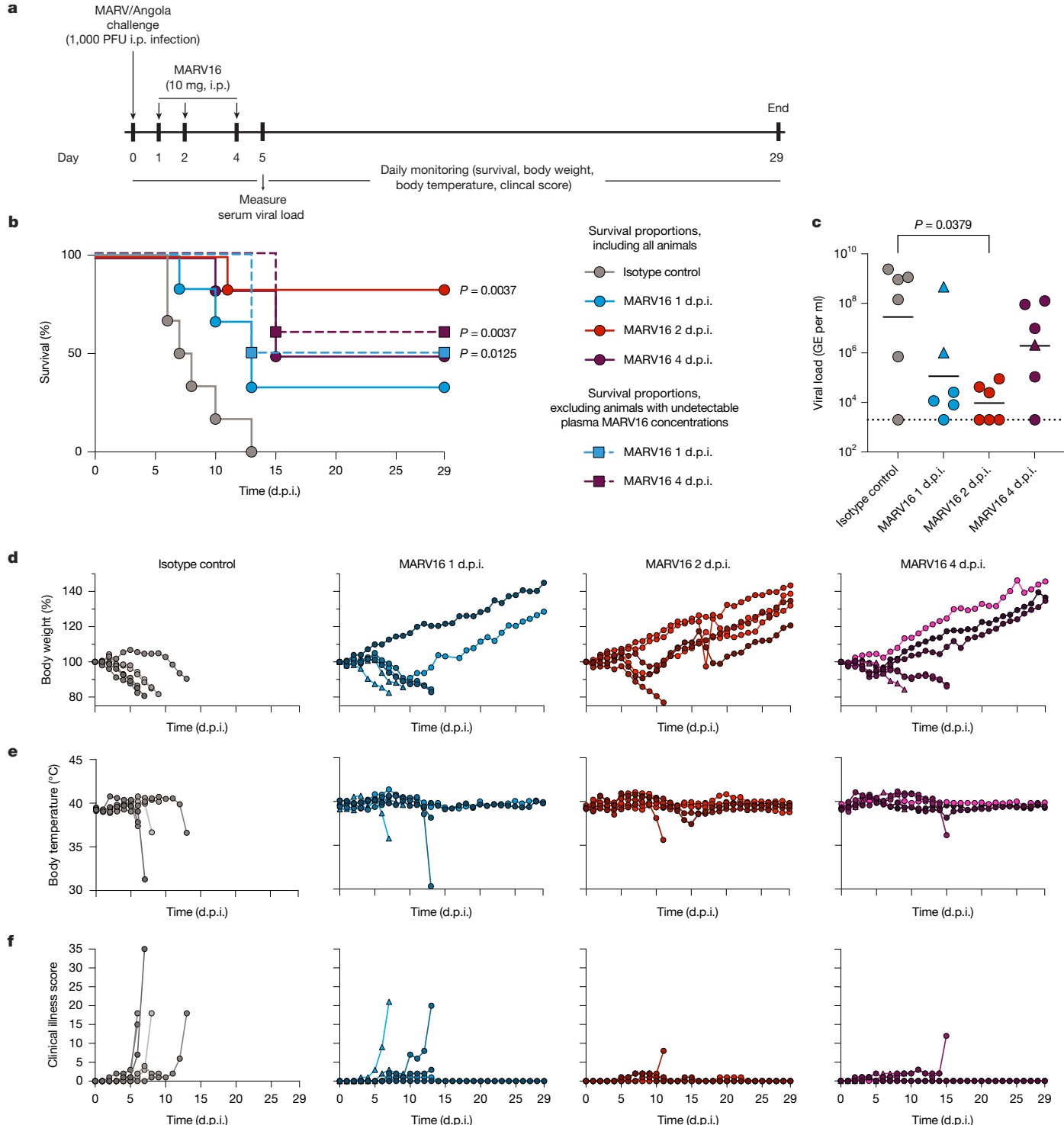

**Fig. 4 | MARV16 protects guinea pigs against MVD. a**, Schematic of the MARV challenge study assessing the therapeutic efficacy of MARV16. i.p., intraperitoneal. **b**, Survival curves for guinea pigs (*n* = 6 per group) challenged with 1,000 PFU of MARV/Angola and treated with MARV16 or an isotype control antibody. Animals were monitored for 29 d.p.i. Survival curves for the groups administered with MARV16 1 d.p.i. and 4 d.p.i. excluding the animals with undetectable plasma MARV16 concentrations 1 day after treatment are shown as boxes with dashed lines. Statistical differences in survival between groups were assessed using Kaplan–Meier survival analysis and excluding animals with undetectable plasma concentrations of MARV16. A two-sided log-rank test was used and Holm–Šídák correction was applied for multiple comparisons between the isotype control antibody-treated guinea pigs and MARV16-treated groups. **c**, MARV/Angola viral loads measured in the plasma of infected guinea pigs at

5 d.p.i. The black line indicates the geometric mean viral load for each group and the dotted line denotes the limit of detection (viral load of ≤2 × 10³ GE per ml). Statistical differences in viral loads between groups were assessed using the Kruskal–Wallis test with Dunn's post-test comparing the isotype control antibody-treated guinea pigs to the MARV16-treated groups (excluding animals with undetectable plasma concentrations of MARV16). **d–f**, Daily body weights (**d**), body temperatures (**e**) and clinical scores (**f**) for guinea pigs for the duration of the challenge study. Animals with undetectable plasma concentrations of MARV16 1 day after treatment are denoted with triangles, which indicates that in these animals, MARV16 was sequestered at the injection site and cleared or 'soaked up' by the challenge virus immediately before reaching the blood stream.

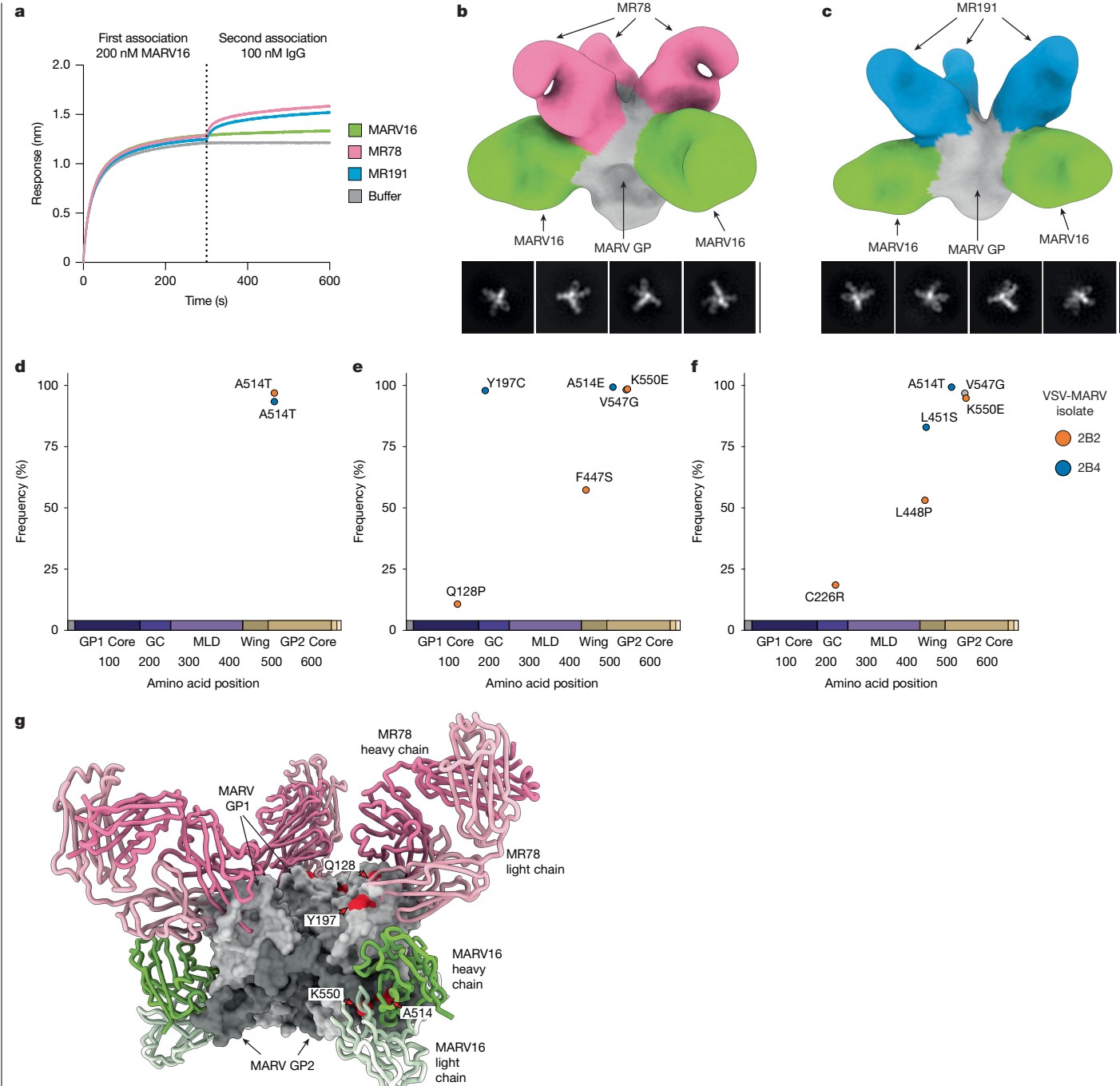

**Fig. 5 | Formulation of MARV antibody cocktails. a**, Competitive binding assay of MARV16, MR78 and MR191 IgG to the MARV16-bound MARV GPΔMuc_WT ectodomain using BLI. Data presented are from one biological replicate and are representative of data from two biological replicates using distinct batches of protein. **b,c**, Representative 2D class averages and 3D reconstruction of negatively stained MARV GPΔMuc_WT ectodomain bound to MR78 and MARV16 Fab fragments (**b**) or MR191 and MARV16 Fab fragments (**c**). The position of the MR78 (pink) or MR191 (blue) Fab fragments were determined by superimposing the RAVV GP–MR78 Fab (PDB: 5UQY) or RAVV GP–MR191 Fab (PDB: 6BP2) structures with our MARV GP–MARV16 Fab structure. Scale bar, 400 Å. **d–f**, Escape mutations identified for MARV16 alone (**d**), the MARV16–MR78 antibody cocktail (**e**) and the MARV16–MR191 antibody cocktail (**f**) using replication-competent VSV encoding the MARV/Musoke GP instead of the VSV G protein. Two selection experiments were performed using the separately plaque-purified VSV-MARV/Musoke GP isolates 2B2 and 2B4. The virus was passaged in the presence of increasing concentrations of antibody until observation of obvious cytopathic effects (>20% of the field of views) in the presence of 100 μg ml⁻¹ of antibody. Mutations were identified by deep sequencing of the viral supernatant, and those that reached a frequency of at least 10% are displayed in the plots and coloured according to the isolate they were identified from. The V547G mutation that was also identified in 2B2 passaged without antibody is displayed in grey. **g**, Escape mutations (red) identified during the antibody-selection experiments mapped on the surface of MARV GPΔMuc_WT (grey). The MR78 position (pink) was determined by superimposing the RAVV GP–MR78 Fab (PDB: 5UQY) structure with our structure of MARV GP in complex with the MARV16 Fab (green). C226, F447S and L448P were not resolved in our structure and are not displayed on the MARV GP.

## Discussion

The identification of stabilizing mutations is a key goal of vaccine design as the use of such mutations can markedly improve the immunogenicity of viral antigens by preferentially eliciting antibodies directed towards the desired conformation of a GP. Such success has been achieved for SARS-CoV-2 S and respiratory syncytial virus F vaccines, which incorporate prefusion-stabilizing mutations[32,33]. The stabilizing mutations identified here improved the expression, thermostability and immunogenicity of MARV GPΔMuc. However, as the serum-neutralizing antibody titres were modest in mice immunized with MARV GPΔMuc$_{WT}$ or MARV GPΔMuc$_{PV}$, strategies such as multimeric presentation of the stabilized MARV GP on nanoparticles[47–49] or inclusion of the stabilizing mutations into mRNA vaccines encoding for the full-length MARV GP[42,50] will probably be necessary for inducing more potent neutralizing antibody titres. We also used ProteinMPNN[45] to identify stabilizing mutations in the MARV GPΔMuc, similar to approaches recently used for the Langya virus G and Epstein–Barr virus gB proteins[43,44], thereby further demonstrating the utility of machine-learning-enabled vaccine design.

Previous studies have suggested that the MARV GP equator and base are shielded from neutralizing antibodies by the GP1 mucin-like domain and the GP2 wing[10,11] given that all previously characterized MARV GP-neutralizing antibodies solely target the RBS[9,36]. As MARV16 binds to an epitope that spans GP1 and GP2, our data showed that the mucin-like domain and wing do not fully shield GP2. Instead, the wing is conformationally flexible, thereby enabling antibody binding. As a result, we anticipate that future antibody discovery campaigns will identify neutralizing antibodies that target multiple different GP2 epitopes. MARV16 recognizes a conserved *Marburgvirus* epitope and neutralizes filoviruses as distantly related as DEHV, which only shares 49.9% amino acid identity with MARV GP. Several *Ebolavirus* GP-directed antibodies that neutralize EBOV, SUDV and Bundibugyo ebolavirus, but not MARV, including ADI-15946, EBOV-515 and EBOV-520, recognize similar epitopes to MARV16 (refs. 52–55), which indicates that this epitope is a prime target for broad genus-specific neutralization. Therapeutics or vaccines that target this MARV GP antigenic site will therefore probably provide robust protection against pre-emergent MARV variants and MARV-related filoviruses, similar to those developed for EBOV[48].

For *Ebolaviruses*, a structured glycan cap blocks access to the RBS until GP cleavage mediated by cathepsin B or L in the endosomes[27,28,51]. This characteristic limits the elicitation of and potency of RBS-directed antibodies[9,10]. By contrast, the glycan cap had not been visualized for MARV GP, and RBS-directed antibodies mediate weak but detectable MARV neutralization, which suggested that the MARV GP RBS is more exposed than that of *Ebolavirus* GPs[9–11,36]. Our structure revealed that the MARV glycan cap shields the RBS in a similar manner to the glycan cap of *Ebolaviruses*. However, the increased conformational heterogeneity or looser tethering of the MARV GP glycan cap to the RBS might enable easier displacement, which explains why MR78 and MR191 neutralize MARV, but not *Ebolaviruses*. Accordingly, we observed an increase in binding for the RBS-directed antibodies MR78 and MR191 to the MARV GPΔMuc$_{WT}$ ectodomain in acidic conditions. These data suggest that the glycan cap is less likely to mask the RBS in the acidic conditions of late endosomes.

As diagnosis and treatment of MVD is often delayed in humans, monoclonal antibodies have been evaluated on the basis of their capacity to provide therapeutic benefits in animal models when administered several days after MARV exposure[39,40]. We showed here that MARV16 provides significant protection against MARV infection in guinea pigs when administered as late as 96 h after infection, which indicates that MARV16 is a promising candidate for treating MVD. Moreover, therapeutic antibody cocktails consisting of multiple antibodies that target distinct epitopes on an antigen are favoured for viral pathogens as the targeted viral protein would typically need to accumulate multiple mutations to evade all the antibodies in the cocktail[57]. All previous neutralizing antibodies identified against the MARV GP target the RBS, which limited the development of an antibody cocktail for MARV[9,36–38]. Our demonstration that MARV16 can bind to the MARV GP concurrently with RBS-directed antibodies indicates that a therapeutic antibody cocktail against MARV, similar to ZMapp for EBOV[20], can be developed. We showed here that such cocktails require multiple GP amino acid substitutions to escape neutralization by both antibodies in the cocktail. These data indicate that antibody cocktails composed of MARV16 and a RBS-directed antibody can be used as MARV therapeutics to provide increased resistance to mutations. Indeed, such combination should retain efficacy during treatment even if the virus evolves.

In summary, our results will inform both vaccine and therapeutic development against MARV, providing improved treatment and prevention options for future MARV outbreaks.

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

## Methods

### Cells

HEK-293T (ATCC), Vero E6 (ATCC), BHK-21/WI-2 (Kerafast) and BS-C-1 cells (ATCC) were grown in DMEM supplemented with 10% fetal bovine serum (FBS) and 1% penicillin–streptomycin–L-glutamine (PS) at 37 °C and 5% $CO_2$. Expi293 (ThermoFisher) and ExpiCHO-S (Thermo-Fisher) were grown in Expi293 or ExpiCHO medium, respectively, at 37 °C and 8% $CO_2$ rotating at 130 rpm. Cell lines were not authenticated or tested for mycoplasma contamination.

### In vivo animal studies

Sixty (30 male and 30 female) 6–8-week-old BALB/c mice (*Mus musculus*) were obtained from Inotiv. Mice were housed at the Bioqual vivarium facility (12-h light–dark cycle, temperature of 20–26.1 °C and relative humidity of 30–70%) with free access to sterilized water and chow. Mice were handled in accordance with the standards of the Association for Assessment and Accreditation of Laboratory Animal Care (AAALAC) International's reference resource: the eighth edition of the 'Guide for the Care and Use of Laboratory Animals', Animal Welfare Act as amended, and the 2015 reprint of the Public Health Service (PHS) Policy on Human Care and Use of Laboratory Animals. All experiments were performed under the Bioqual Institutional Animal Care and Use Committee (IACUC)-approved protocol number 23-054P. Animals were inspected before inclusion in experiments and monitored throughout the study by a veterinarian.

ATX-GK and ATX-GL female mice, 6–7 weeks old, were obtained from Alloy Therapeutics and housed for the immunization experiment at the Institute for Research in Biomedicine, Bellinzona, Switzerland. All animal experiments were performed in accordance with the Swiss Federal Veterinary Office guidelines and authorized by the Cantonal Veterinary (approval number 35554 TI-39/2023/2023). Animals were supervised by a licensed veterinarian, and proper steps were taken to ensure the welfare and to minimize the suffering of all animals in the studies. Animals were housed in ventilated cages in a 12-h light–dark cycle at 20 ± 2 °C and a relative humidity of 55 ± 8%, with free access to water and standard sterilized chow.

Twenty-four female, 4–8-week-old Hartley guinea pigs (*Cavia porcellus*) were obtained from Charles River Laboratories and housed in the Centers for Disease Control and Prevention (CDC)-accredited Biosafety Level 4 (BSL-4)/Animal Biosafety Level 4 (ABSL-4) containment facility at Texas Biomed (temperature of 18–28 °C and relative humidity of 25–75%) with free access to water and chow. All experiments were approved by Texas Biomed's IACUC under protocol 1915C before the initiation of the study and performed in accordance with the Animal Welfare Act and the Guide for the Care and Use of Laboratory Animals. Animals were observed by a veterinarian at least once daily before exposure to the challenge virus and at least twice daily after administration of the challenge virus. Moribund animals were euthanized with the approval of a veterinarian.

### Constructs

The construct encoding the MARV GPΔMuc$_{WT}$ ectodomain (residues 1–256 and 426–637) with a carboxy-terminal 8× His tag was codon-optimized, synthesized and inserted into pcDNA3.1(+) by Genscript. Mutations neighbouring the furin-cleavage site (W439A, F445G and F447N) and the stabilizing mutation (H589I) were introduced using In-Fusion Cloning with overlapping mutagenesis primers. The HR1$_c$-stabilizing mutations (T582P and F583V) were also introduced through In-Fusion Cloning using overlapping mutagenesis primers. Constructs encoding the EBOV GPΔMuc domain (residues 1–312 and 463–637) and the SUDV GPΔMuc domain (residues 1–313 and 474–637) both with a C-terminal T4 foldon and 8× His tag were codon-optimized, synthesized and inserted into pcDNA3.4 by Genscript. The constructs encoding the full-length MARV/Musoke GP

(GenBank accession number: NC_001608) with a C-terminal Flag tag was codon-optimized, synthesized and inserted into pcDNA3.1(+) by Genscript. Constructs encoding the full-length MARV/Ci67 (GenBank accession: EF446132), MARV/Ozolin (GenBank accession: AY358025), MARV/Angola (GenBank accession: KY047763), MARV/Kakbat-SL-2017 (GenBank accession: MN258361), MARV/Kasbat-SL-2018 (GenBank accession: MN187403), MARV/Ghana-2022 (GenBank accession: OQ672470) and MARV/Equatorial Guinea-2023 (HS415030) were codon-optimized, synthesized and inserted into pHDM by Genscript. Constructs encoding furin and the MR78, MR191 and EBOV515 heavy and light chains were codon-optimized, synthesized and inserted into pcDNA3.1(+) by Genscript. Constructs encoding the MARV4, MARV7, MARV11, MARV14, MARV16, MARV18, MARV20, MARV21 and MARV23 heavy and light chains were generated by cloning variable regions into IgG1 and IgK expression vectors[58]. VH and VL amino acid sequences for the ten MARV antibodies are presented in Extended Data Table 1. Constructs encoding the MARV7, MARV14, MARV18 and MARV20 VH-CH1 sequence with an N-terminal CD5 leader sequence and C-terminal His tag were codon-optimized, synthesized and inserted into pcDNA3.4 by Genscript. Constructs encoding the MARV7, MARV14, MARV18 and MARV20 VL-CH1 sequence with an N-terminal CD5 leader sequence and C-terminal 3× Flag tag were codon-optimized, synthesized and inserted into pcDNA3.4 by Genscript. The native, full-length MARV/Musoke GP was synthesized and inserted between the VSV M and L proteins in pVSV eGFP dG (a gift from C. Cepko; Addgene plasmid 31842) by Genscript. Helper plasmids encoding the VSV N, P, L and G proteins were obtained from Kerafast (EH1012).

### Recombinant protein expression and purification

To produce the MARV GPΔMuc, EBOV GPΔMuc and SUDV GPΔMuc ectodomains, Expi293 cells were grown to a density of $3 × 10^6$ cells per ml and then transfected with constructs encoding the ectodomain and furin at a 3:1 mass ratio using an Expifectamine293 transfection kit following the manufacturer's instructions. Five days after transfection, the supernatant was collected, clarified by centrifugation and incubated with Ni Sepharose Excel resin (Cytiva) for 1 h at room temperature. The resin was then collected in a gravity column and washed with buffer containing 25 mM sodium phosphate pH 8.0, 300 mM NaCl and 50 mM imidazole or 100 mM Tris pH 8.0, 300 mM NaCl and 40 mM imidazole. The proteins were then eluted using an elution buffer containing 25 mM sodium phosphate, 300 mM NaCl, 500 mM imidazole, pH 8.0 or 100 mM Tris, 300 mM NaCl, 300 mM imidazole and further purified into TBS (20 mM Tris pH 7.4 and 100 mM NaCl, or 50 mM Tris pH 7.4 and 150 mM NaCl) by size-exclusion chromatography using a Superose 6 Increase 10/300 GL column. The purified proteins were concentrated using a 100 kDa Amicon centrifugal filter, flash frozen and stored at −80 °C until use.

MARV7, MARV11, MARV14, MARV16, MARV18, MARV20, MARV21 and MARV23 were expressed recombinantly by transient transfection of ExpiCHO-S cells (ThermoFisher Scientific) using an ExpiFectamine CHO transfection kit (Thermo Fisher Scientific). After 8 days, cell culture supernatant was separated with a Sartoclear Dynamics Lab V kit (Sartorius) and affinity purified by protein A chromatography using ÄKTA Xpress Fast Protein liquid chromatography (Cytiva) with HiTrap protein A columns (Cytiva) followed by buffer exchange to a buffer containing 20 mM histidine and 150 mM NaCl, pH 6.0 using HiPrep 26/10 desalting columns (Cytiva). The purified antibody concentrate was quantified using a Lunatic spectrophotometer (Unchained Labs) and stored at −80 °C until use.

MR78, MR191 and EBOV515 monoclonal antibodies were produced by transfecting Expi293 cells grown to a density of $3 × 10^6$ cells per ml with the heavy chain and light chain constructs supplied at a 1:1 mass ratio using an Expifectamine293 transfection kit. Four to 5 days after transfection, the supernatant was collected, clarified by centrifugation and flowed over a protein A column. The column was then washed with

at least ten column volumes of wash buffer containing 20 mM sodium phosphate, pH 8.0. The eluted antibodies were then exchanged into TBS and concentrated using a 100 kDa Amicon centrifugal filter.

MARV7, MARV14, MARV18 and MARV20 Fab fragments were produced by transfecting Expi293 cells with the heavy and light chain constructs at a 1:1 mass ratio using an Expifectamine293 transfection kit and following the manufacturer's recommendations. Four to 5 days after transfection, the supernatant was collected, clarified by centrifugation and incubated with Ni Sepharose Excel resin (Cytiva) for 1 h at room temperature. The resin was collected in a gravity column and washed with buffer containing 25 mM sodium phosphate pH 8.0, 300 mM NaCl and 20 mM imidazole. The proteins were then eluted using elution buffer containing 25 mM sodium phosphate, 300 mM NaCl, 500 mM imidazole, pH 8.0, and further purified into TBS (20 mM Tris pH 7.4 and 100 mM NaCl) by size-exclusion chromatography using a Superdex 75 Increase 10/300 GL column, and concentrated using a 30 kDa Amicon centrifugal filter.

### Biotinylation of MARV GPΔMuc$_{WT}$
The MARV GPΔMuc$_{WT}$ was produced and purified as described above. After elution from Ni Sepharose Excel resin, the GP was exchanged into PBS (137 mM NaCl, 2.7 mM KCl, 10 mM Na$_2$HPO$_4$ and 1.8 mM KH$_2$PO$_4$, pH 7.4) and concentrated to 1 mg ml$^{-1}$ using a 100 kDa Amicon centrifugal filter. The MARV GPΔMuc$_{WT}$ was biotinylated using an EZ-Link Sulfo-NHS-SS-Biotinylation kit (ThermoFisher) with a 40-fold molar excess of Sulfo-NHS-SS-biotin and incubating the reaction mixture at room temperature for 30 min. The biotinylated MARV GPΔMuc$_{WT}$ was then purified into TBS by size-exclusion chromatography using a Superose 6 Increase 10/300 GL column. The purified protein was then concentrated using a 100 kDa Amicon centrifugal filter, flash frozen and stored at −80 °C until use.

### Cleavage of EBOV and SUDV GPΔMuc
The EBOV and SUDV GPΔMuc ectodomains were produced and purified as described above. After elution from Ni Sepharose Excel resin, the GP was exchanged into TBS and concentrated to 1 mg ml$^{-1}$ using a 100 kDa Amicon centrifugal filter. Thermolysin (Sigma-Aldrich), resuspended in TBS, was added to a final concentration of 0.2 mg ml$^{-1}$. The reaction mixture was incubated at 37 °C for 1 h, after which phosphoramidon (Sigma-Aldrich) was added to a final concentration of 500 μM to stop thermolysin. The cleaved GP was purified into TBS by size-exclusion chromatography using a Superdex 200 Increase 10/300 GL column.

### Generation of Fab fragments
To generate Fab fragments from the purified monoclonal antibodies, LysC, resuspended in TBS, was added to 1 mg of MARV16, MR78 or MR191 IgG at 1:4,000 to 1:8,000 mass ratios and incubated at 37 °C overnight. The following day, Cytiva MabSelect resin was added to the digested IgG solution and incubated for 1 h at room temperature. The flow through was collected and run over a Superdex 75 Increase 10/300 GL column equilibrated into TBS. Fractions containing the Fab fragment were pooled and concentrated using a 30 kDa Amicon centrifugal filter.

### Immunogenicity study with BALB/c mice
Mice were randomized to form six groups ($n$ = 10 mice per group) with equal numbers of male and female mice. On study days 0, 28 and 56, 0.1 μg, 1 μg or 10 μg of MARV GPΔMuc$_{WT}$ or MARV GPΔMuc$_{PV}$ diluted in TBS was mixed with InvivoGen Addavax at a 1:1 volume ratio for a total injection volume of 100 μl. The mice were intramuscularly immunized in both quadriceps (50 μl per quadricep). On study days −7, 14, 42 and 70, blood was collected from each animal in serum-separator tubes. The resulting serum was stored at −80 °C until use.

### MARV GPΔMuc immunizations and antibody discovery from Alloy mice
Pre-immune serum was obtained from each mouse 1 week before immunization. ATX mice were immunized with recombinant MARV GPΔMuc$_{WT}$ diluted (1:1) in Magic Mouse adjuvant (CDN-A001E; CD Creative Diagnostics) and subcutaneously and intraperitoneally injected. On day 0, mice received prime immunization with 20 μg of MARV GPΔMuc$_{WT}$ and were boosted on day 13 and day 64 with the same amount of antigen. On day 70, the mice were killed and peripheral blood, spleen and lymph nodes were collected and cells were freshly isolated. B cells from either freshly isolated or frozen splenocytes were enriched by positive selection using mouse CD19 microbeads and LS columns (Miltenyi) and subsequently stained with mouse anti-IgM (BioLegend, 406508; 2 μg ml$^{-1}$), anti-IgD (BioLegend, 405706; 2 μg ml$^{-1}$), anti-IgA (Fisher Scientific, 15208769; 2 μg ml$^{-1}$) and biotinylated MARV GPΔMuc$_{WT}$ labelled with both streptavidin-Alexa-Fluor 488 and streptavidin-Alexa-Fluor 647 (Life Technologies). MARV GPΔMuc$_{WT}$-specific IgG$^+$ memory B cells were sorted by flow cytometry by gating out IgM$^+$IgD$^+$IgA$^+$ B cells and positively baiting B cells with dual-labelled (Alexa-Fluor 488 and Alexa-Fluor 647) antigen using a SH800SFP cell sorter (Sony). Sorted IgG$^+$ memory B cells were seeded at clonal dilution in 384-well plates on a monolayer of feeder mesenchymal cells in the presence of B cell survival factors. Clones positive for antigen binding were then isolated and cDNA was synthesized. Monoclonal antibody VH and VL sequences were obtained by RT–PCR. Genes encoding V, D and J of the IgH DNA sequences were identified using the IMGT database as a ref. 59. Monoclonal antibodies were then produced recombinantly as human IgG1 (IgG1m3 allotype) in ExpiCHO cells and transiently transfected with heavy and light expression vectors as previously described[58].

### ELISA
The MARV GPΔMuc$_{WT}$ or MARV GPΔMuc$_{PV}$ ectodomain was diluted to 0.003 mg ml$^{-1}$ in TBS, added to Maxisorp 384-well plates (ThermoFisher) and incubated overnight at room temperature. The following day, the plates were tapped until dry and blocked with blocker casein for 1 h at 37 °C. The plates were tapped dry, and serum samples (starting concentration of 1:40) or monoclonal antibodies (starting concentration of 0.1 mg ml$^{-1}$) were diluted in TBS supplemented with 0.1% Tween 20 (TBST) and serially diluted 1:3 in TBST, added to plates and incubated at 37 °C for 1 h. The plates were tapped dry and washed 4 times with TBST, after which a goat anti-mouse IgG (H+L) HRP-conjugated antibody (ThermoFisher) diluted 1:5,000 in TBST or a goat anti-human IgG (H+L) HRP-conjugated antibody (ThermoFisher) diluted 1:5,000 in TBST was added to each well. The plates were incubated for 1 h at 37 °C, tapped dry and washed 4 times with TBST. SureBlue Reserve TMB 1-Component Microwell Peroxidase substrate (SeraCare) was added to each well and developed for 90 s at room temperature. An equal volume of 1 N HCl was added to each well to quench the reaction, after which the absorbance at 450 nm was measured using a BioTek Synergy Neo2 plate reader. The resulting data were analysed using GraphPad Prism 10, using a four-parameter logistic curve to determine the ED$_{50}$ for each antibody. Two biological replicates performed in technical duplicate were performed using two distinct batches of protein.

### Pseudotyped virus production
VSV pseudotyped with the full-length MARV, RAVV, DEHV, MLAV, EBOV or SUDV GP was produced as previously described[60–63]. In brief, HEK-293T cells were split into 10-cm poly-lysine-coated dishes and grown overnight at 37 °C and 5% CO$_2$ until they reach approximately 90–95% confluency. The cells were washed once with DMEM and left in DMEM supplemented with 10% FBS. The cells were transfected with 16–24 μg full-length GP construct using Lipofectamine 2000 following the manufacturer's recommendations, after which the cells were

incubated for 20–24 h at 37 °C and 5% $CO_2$. The cells were then washed 3 times with DMEM, infected with VSVΔG/luc and incubated at 37 °C and 5% $CO_2$. After 2 h, the cells were washed 5 times with DMEM and left in DMEM supplemented with an anti-VSV-G antibody (I1-mouse hybridoma supernatant diluted 1:25 for CRL-2700, American Type Culture Collection) for 20–24 h at 37 °C and 5% $CO_2$. Following this incubation, the supernatant was collected, clarified by centrifugation, filtered using a 0.45 μM filter and concentrated with a 100 kDa centrifugal filter (Amicon). The resulting pseudovirus was stored at −80 °C until use.

## Pseudovirus neutralization assay

Neutralization assays were performed as previously described[60–63]. In brief, Vero E6 cells were split into white-walled, clear-bottom 96-well plates at a density of 18,000 cells per well. The cells were grown overnight at 37 °C and 5% $CO_2$ until they reached approximately 80–90% confluency. The serum samples (starting concentration of 1:10) or monoclonal antibodies (starting concentration of 200 μg ml$^{-1}$) were diluted in DMEM and serially diluted 1:3 in DMEM. VSV pseudotyped with the GP was diluted 1:5 to 1:250 in DMEM, after which an equal volume of diluted pseudovirus was added to the diluted monoclonal antibody or serum. The pseudovirus–antibody mixture was incubated at room temperature for 30 min. Following this incubation, growth medium was removed from the Vero E6 cells and the pseudovirus–antibody mixture was added to cells. The cells were incubated for 2 h at 37 °C and 5% $CO_2$, after which an equal volume of DMEM supplemented with 20% FBS and 2% PS was added to each well and the cells were incubated for another 20–24 h at 37 °C and 5% $CO_2$. An equal volume of ONE-Glo EX (Promega) was added to each well and the cells were incubated 37 °C for 5 min with constant shaking. The luminescence values from each well were measured using a BioTek Synergy Neo2 plate reader.

Data were normalized using GraphPad Prism 10 using the relative light unit (RLU) values obtained from uninfected cells to define 100% neutralization and the RLU values obtained from cells infected with pseudovirus in the absence of antibody to define 0% neutralization. $ED_{50}$ values were determined from the normalized data using an [inhibitor] versus normalized response–variable slope model. At least two biological replicates using distinct batches of pseudoviruses and monoclonal antibodies were performed.

## Plaque reduction neutralization assay with authentic MARV

Vero E6 cells were split into 6-well plates at a density of $3 \times 10^5$ cells per well and grown overnight in high-glucose DMEM supplemented with 10% FBS and 1% PS at 37 °C and 5% $CO_2$ until they reached 75–95% confluency. The following day, monoclonal antibodies were diluted in high-glucose DMEM with 2% FBS and 1% PS (DMEM-2) to a starting concentration of 100 μg ml$^{-1}$ and serially diluted 1:4 in DMEM-2. Next, 100 μl of MARV/Musoke diluted in DMEM-2 to 1,000 PFU per ml was added to 100 μl of the diluted antibodies and the virus–antibody mixture was incubated for 60 min at 37 °C. Following this incubation, an additional 300 μl of DMEM was added to the virus–antibody mixture and 400 μl of this mixture was added to the Vero E6 cells. The cells were incubated with the virus–antibody mixture for 60 min at 37 °C with gentle rocking, after which the mixture was removed and the primary overlay consisting of 1% agarose mixed 1:1 with 2× MEM containing 2 mM sodium pyruvate, 1% PS and 4% FBS was added. The cells were incubated for 7 days at 37 °C with 5% $CO_2$. Next, the cells were stained with an overlay containing 1% agarose mixed 1:1 with 2× MEM containing 2 mM sodium pyruvate, 4% FBS and 8% neutral red solution and incubated for 1 day at 37 °C and 5% $CO_2$, after which the number of plaques were manually counted. The per cent infectivity for each well was determined by dividing the number of plaques in the well by the number of plaques counted in the well with 24 pg ml$^{-1}$ of antibody. Two biological replicates with one to three technical replicates were conducted for each antibody, and the $PRNT_{50}$ values were determined from the averaged data from

the two biological replicates using an [inhibitor] versus normalized response–variable slope model in GraphPad Prism 10.

## BLI

Binding of the stabilized MARV GPΔMuc$_{WT}$ ectodomains to MR191 was assessed by first dipping pre-hydrated AHC2 biosensors into MR191 IgG diluted to 10 ng μl$^{-1}$ in 10x kinetics buffer to a 1 nm shift. The MR191-coated biosensors were then dipped into each MARV GPΔMuc construct diluted to 10 nM in 10x kinetics buffer for 300 s, after which the biosensors were dipped into 10x kinetics buffer. All steps were conducted at 30 °C. Data were baseline subtracted using Octet Data Analysis HT software (v.12.0) and visualized using GraphPad Prism 10.

To measure the affinity of the MARV16 Fab for MARV GPΔMuc$_{WT}$, biotinylated MARV GPΔMuc$_{WT}$ was diluted to a concentration of 10 ng μl$^{-1}$ in 10x kinetics buffers and loaded onto pre-hydrated streptavidin biosensor to a 1 nm shift. The MARV GPΔMuc$_{WT}$-coated biosensors were then dipped for 300 s into MARV16 Fab diluted in 10x kinetics buffer at a starting concentration of 100 nM and serially diluted 1:3. The biosensors were then dipped into 10x kinetics buffer for an additional 300 s. All steps were conducted at 30 °C. The resulting data were baseline subtracted and fit using Octet Data Analysis HT software (v.12.0) and visualized using GraphPad Prism 10.

Binding comparisons between the MARV16, MR78 and MR191 Fab and IgG fragments were conducted as described above. Following immobilization of the biotinylated MARV GPΔMuc$_{WT}$ on the streptavidin biosensors, the tips were dipped for 300 s into 100 nM of Fab or IgG diluted in 10x buffer, after which the tips were dipped into 10x kinetics buffer for 300 s. All steps were conducted at 30 °C. Data were baseline subtracted using Octet Data Analysis HT software (v.12.0) and visualized using GraphPad Prism 10.

Binding of MARV GPΔMuc to MARV16, MR78 or MR191 IgG at variable pH values were conducted by loading biotinylated MARV GPΔMuc$_{WT}$ diluted in 10x kinetics buffer, pH 7.4 onto streptavidin biosensors to a 1 nm shift. The loaded biosensors were then dipped for 300 s into IgG diluted in 10x kinetics buffer at pH 7.4, 6.5, 5.5 or 4.5. The resulting data were baseline subtracted using Octet Data Analysis HT software (v.12.0) and visualized using GraphPad Prism 10.

Binding of the MARV GPΔMuc$_{WT}$, EBOV GPΔMuc, cleaved EBOV GP, SUDV GPΔMuc and SUDV GPΔMuc to MARV16, MR78, MR191 and EBOV515 IgG was assessed as described above. IgG diluted to 10 ng μl$^{-1}$ in 10x kinetics buffer was loaded on AHC2 biosensors to a 1 nm shift, after which the loaded biosensors were dipped for 300 s into GP diluted to approximately 10 nM in 10x kinetics buffer. All steps were conducted at 30 °C. Data were baseline subtracted using Octet Data Analysis HT software (v.12.0) and visualized using GraphPad Prism 10.

Competitive binding of MARV16 versus MR78 or MR191 to the MARV GPΔMuc$_{WT}$ was assessed by loading biotinylated MARV GPΔMuc$_{WT}$ diluted to 10 ng μl$^{-1}$ onto pre-hydrated streptavidin biosensor, after which the loaded biosensors were dipped for 300 s into 200 nM of MARV16 IgG diluted 10x kinetics buffer. The biosensors were then dipped into 100 nM of MR78, MR191 or MARV16 IgG diluted in 10x kinetics buffer for 300 s and finally dipped into 10x kinetics buffer for 300 s. All steps were conducted at 30 °C. The resulting data were baseline subtracted using Octet Data Analysis HT software (v.12.0) and visualized using GraphPad Prism 10.

For epitope binding of the MARV antibodies discovered from Alloy ATX mice, biotinylated MARV GPΔMuc$_{WT}$ was diluted to 10 ng μl$^{-1}$ and loaded onto pre-hydrated streptavidin biosensors to a 1 nm shift. The loaded biosensors were dipped into 200 nM of the saturating antibody diluted in 10x kinetics buffer for 900 s and then dipped into 100 nM of the competing antibody and 25 nM of the saturating antibody diluted in 10x kinetics buffer for 300 s. All steps were conducted at 30 °C. The resulting data were analysed with Octet Data Analysis HT software (v.12.0), in which the response for the saturating antibody was calculated by subtracting the average response measured for the last 30 s

of the association step of the saturating antibody from the average response measured for the last 90 s of association step of the competing antibody. The resulting data were corrected by subtracting the response measured for self-blocking and the per cent binding was calculated by dividing the response for each competing–saturating antibody pair by the response measured for the saturating antibody binding to the MARV GPΔMuc_{WT} alone. Antibody pairs displaying reciprocal blocking relationships were considered to be a part of the same binding group.

## In vivo challenge study

Twenty-four guinea pigs were randomly assigned to four groups ($n = 6$ animals per group). The animals were intraperitoneally infected with 1,000 PFU of guinea pig-adapted MARV/Angola diluted in 100 µl PBS. Twenty-four, 48 or 96 h after infection, the animals were intraperitoneally treated with 10 mg (approximately 20 mg kg$^{-1}$) of MARV16 or an isotype control monoclonal antibody (MGH2). Blood samples were collected 24 h after administration of the antibody and 5 d.p.i. Body weights, body temperatures and clinical illness scores were recorded daily until the end of the study (29 d.p.i.). Clinical illness scores were assigned as follows. Weight loss: 0, decrease from baseline body weight between 0 and 4.99%; 1, decrease from baseline body weight equal or higher than 5 and less than 10.9%; 2, decrease from baseline body weight equal or higher than 11 and less than 19.9%; and 3, decrease from baseline body weight equal or higher than 20%. Temperature changes: 0, no change from baseline; 1, a change equal or higher than 1.1 °F; and 3, a change equal or higher than 2.2 °C. Dyspnoea: 0, normal respiration; 3, rapid respiration; and 12, laboured or agonal respiration. Responsiveness: 0, active; 2, mild unresponsiveness, becomes active when approached; 8, moderate unresponsiveness, lethargic, weakness; and 15, moribund or prostrate. Discoordination: 0, none; or 2, noticeable. Appearance: 0, active and alert; 1, rough hair coat; and 3, rough hair coat and hunched. Eye appearance: 0, normal; 1, discharge from eye; 2, squinty eye (or eyes); and 3, closed eyes. The total clinical illness scores were determined by adding up all clinical scores from the aforementioned categories. When animals reached euthanasia criteria (total clinical illness scores of 12–35), they were euthanized with the approval by the study's veterinarian.

To measure MARV viral loads at 5 d.p.i., 100 µl of plasma collected from each animal was mixed with 150 µl PBS and inactivated with 750 µl TRIzol LS reagent. Next, 10 µg yeast tRNA and 10$^3$ PFU of MS2 bacteriophage were added to the sample. Next, 200 µl chloroform was added to each sample and the samples were centrifuged at 12,000$g$ for 15 min at 2–8 °C. The aqueous phase was transferred to a Microtiter Deep-well 96 plate and RNA was extracted using a NucleoMag Pathogen kit (Macherey-Nagel) with a KingFisher Flex instrument. The extracted RNA was stored at −80 °C until use. RT–qPCR was performed using TaqPath 1-Step RT–qPCR master mix (ThermoFisher) using primers and a probe targeting the GP gene of MARV/Angola (MAGP forward primer: CCAAACGATGGGCCTTCA; MAGP reverse primer: TCCT CCCCTTCTGTATACTCAACAT; MAGP FAM/MGB probe: CAGGTGTA CCTCCCC). A standard curve was generated using a 1:10 serially diluted ssRNA standard (10$^7$ to 10$^1$ copies per 5 µl) and a MS2 phage assay was conducted as an internal extraction and detection control. Two technical replicates were conducted for each sample. Results are expressed as GE per ml of plasma.

To measure plasma concentrations of MARV16, plasma was collected from guinea pigs 1 day after antibody administration. MARV GPΔMuc_{WT} was diluted in PBS to 2 µg ml$^{-1}$, added to Immulon 2 HB 96-well flat-bottom plates (ThermoFisher) and incubated overnight at 4 °C. The following day, the plates were washed 3 times with PBST, blocked for 1 h at 2–8 °C using Pierce Protein-Free (PBS) blocking buffer (ThermoFisher) and then washed 3 more times with PBST. Next, plasma samples diluted 1:10, 1:400 and 1:2,000 in PBS with 1% FBS and 0.2% Tween 20 were added to the wells and the plates were incubated

for 1 h at 37 °C. The plates were washed 3 times with PBST and a goat anti-human IgG-HRP-conjugated secondary antibody (Millipore Sigma) diluted 1:6,000 in PBS with 1% FBS and 0.2% Tween 20 was added to each well. The plates were incubated 1 h at 37 °C and washed 3 times with PBST. TMB substrate (ThermoFisher) was added to each well, developed for 12 min and stopped with an equal volume of 2 N H$_2$SO$_4$. Optical densities at 450 nm were measured using a BioTek 800 TS spectrophotometer. A standard curve was generated using MARV16 diluted in PBS to 200 ng ml$^{-1}$ and 1:2 serially diluted 7 times. The standard curve was fitted with a four-parameter logistic curve and used to calculate plasma concentrations of MARV16. Two technical replicates were conducted for each sample.

## FcγRIIa or FcγRIIIa activation assays

Antibody-dependent activation of human FcγRIIIa and FcγRIIa was performed with a bioluminescent reporter assay. ExpiCHO cells transiently expressing membrane-anchored wild-type MARV/Musoke GP (target cells) were incubated with different amounts of monoclonal antibodies. After 25 min, Jurkat cells stably expressing FcγRIIIa receptor (V158 variant) or FcγRIIa receptor (H131 variant) and NFAT-driven luciferase gene (effector cells) were added at an effector to target ratio of 6:1 for FcγRIIIa assays and 5:1 for FcγRIIa assays. Signalling was quantified by the luciferase signal produced as a result of NFAT pathway activation. Luminescence was measured after 22 h of incubation at 37 °C with 5% CO$_2$ with a luminometer using Bio-Glo-TM luciferase assay reagent according to the manufacturer's instructions (Promega).

## Cryo-EM sample preparation and data collection

To generate MARV GPΔMuc_{WT}–MARV16 Fab complexes, 100 µg of MARV GPΔMuc_{WT} and 150 µg MARV16 Fab were incubated together at 37 °C for 30 min, after which the complexes were added to a 100 kDa centrifugal filter, washed 5 times with TBS and concentrated to 5.5 mg ml$^{-1}$. Immediately before freezing, CHAPSO was added to the MARV GPΔMuc_{WT}–MARV16 Fab complexes to a final concentration of 5.45 mM. The complexes were added to freshly glow-discharged 2.0/2.0 UltraFoil grids (200 mesh)[64], after which the grids were plunge frozen using a Vitrobot MarkIV (ThermoFisher) with a wait time of 10 s, a blot force of 0 and a blot time of 5 s at 100% humidity and 23 °C. Data were acquired on a FEI Titan Krios transmission electron microscope operated at 300 kV and equipped with a Gatan K3 direct detector and Gatan Quantum GIF energy filter, operated in zero-loss mode with a slit width of 20 eV. Automated data acquisition was carried out using Leginon[65] at a nominal magnification of ×105,000 with a pixel size of 0.829 Å, a defocus range between −0.4 to −3.0 µm and a stage tilt of 0° or 25°. The dose rate was adjusted to 15 counts per pixel per s and each video was acquired in 75 frames of 40 s.

## Cryo-EM data processing

Video frame alignment with a downsampled pixel size of 1.658 Å was carried out in WARP[66]. Estimation of microscope CTF parameters, particle picking and extraction (box size of 256 pixels$^2$) was conducted using cryoSPARC (v.4.6.2). Reference-free 2D classification to select well-defined particle images was performed in cryoSPARC[67]. Next, ab initio 3D reconstruction and heterogeneous refinement to select well-defined particle classes were performed in cryoSPARC. 3D refinements were then conducted using nonuniform refinement[68] with C3 symmetry and per-particle defocus refinement in cryoSPARC[68]. The particle images were then subjected to Bayesian polishing using Relion[69], during which the box size was adjusted to 512 pixels$^2$ and the pixel size was adjusted to 0.829 Å. Another round of nonuniform refinement with global and per-particle defocus refinement was performed in cryoSPARC. Next, focused 3D classification without particle alignment was conducted using a mask comprising GP residues 171–219 using a tau factor of 10 in Relion[70,71]. The particles with the best resolved local density were selected and subjected to local refinement with C3 symmetry

in cryoSPARC using a mask comprising MARV GP and the Fab variable domains. Reported resolutions are based on the gold-standard Fourier shell correlation of 0.143 criterion, and Fourier shell correlation curves were corrected for the effects of soft masking by high-resolution noise substitution[72,73].

## Model building and refinement

USCF ChimeraX[74] and Coot[75] were used to fit into the map initial models of the MARV GP (PDB identifier: 6BP2) and MARV16 Fab, which was predicted using AlphaFold2 (ref. 76). The model was then built and refined into the map using Coot, Rosetta[77,78], ISOLDE[79] and Phenix[80]. Figures were generated using UCSF ChimeraX.

## Differential scanning fluorimetry

The original and stabilized MARV GPΔMuc ectodomains were diluted in TBS and mixed with protein thermal shift buffer and dye (Thermo-Fisher) following the manufacturer's recommendation such that the final concentration of protein in the reaction mix was 0.25 μg ml$^{-1}$. The reaction mix was added to a 96-well qPCR plate (ThermoFisher) and sealed with MicroAmp optical adhesive film (ThermoFisher). The fluorescence intensity ($\lambda_{\text{Excitation}}$: 470 ± 15 nm; $\lambda_{\text{Emission}}$: 586 ± 10 nm) was measured from 25 °C to 99 °C using a QuantStudio 5 Real-Time PCR system (ThermoFisher). The data were analysed and visualized using QuantStudio Design and Analysis Desktop software (ThermoFisher) and GraphPad Prism 10. Data are presented as the negative first derivative of fluorescence intensity with respect to temperature. The $T_m$ was identified by taking the second derivative of the fluorescence intensity with respect to temperature and smoothing the resulting function across four neighbours points. Four biological replicates each with six technical replicates were performed using distinct batches of protein.

## Negative-stain electron microscopy

Complexes of the MARV GPΔMuc$_{WT}$–MARV16 Fab-MR78 Fab, MARV GPΔMuc$_{WT}$–MARV16 Fab–MR191 Fab, MARV GPΔMuc$_{WT}$–MARV7 Fab, MARV GPΔMuc$_{WT}$–MARV14 Fab, MARV GPΔMuc$_{WT}$–MARV18 Fab and MARV GPΔMuc$_{WT}$–MARV20 Fab were generated as described above. Purified MARV GPΔMuc mutants or MARV GPΔMuc$_{WT}$–Fab complexes were diluted to 0.01 mg ml$^{-1}$ in TBS, added to freshly glow-discharged carbon-coated copper grids and stained with 2% uranyl formate. Data were acquired with a 120 kV FEI Tecnai G2 Spirit with a Gatan Ultrascan 4000 4k × 4k CCD camera at a nominal magnification of ×67,000 using Leginon[65]. The defocus ranged from −3.0 to −1.0 μm and the pixel size was 1.6 Å. Micrographs were then processed in cryoSPARC[67] using PatchCTF to estimate microscope CTF parameters and Blob picker to pick particles. Following particle extraction, reference-free 2D classification was performed to select well-defined particle images. Ab initio 3D reconstruction and homogenous refinement were then performed with the selected particle images applying C3 symmetry. Figures were generated using UCSF ChimeraX[74].

## Antibody escape studies using replication-competent VSV-MARV/Musoke GP

These experiments underwent evaluation by the biosafety committee of the University of Washington before approval. Replication-competent VSV-MARV/Musoke GP (lacking VSV G) was generated as previously described[81] but with several modifications. BHK-21/WI-2 cells were split into a 6-well plate and grown overnight at 37 °C and 5% $CO_2$ until they reached approximately 90% confluency. The cells were then infected with vaccinia virus strain vTF7-3 (American Type Culture Collection, VR-2153) at a multiplicity of infection of about 3 for 45 min, after which the virus was removed from the cells and fresh growth medium (DMEM, 10% FBS and 1% PS) was added. The cells were transfected with the VSV-MARV/Musoke GP anti-genome, VSV N, VSV P, VSV L and VSV G constructs at a 1:3:5:1:8 mass ratio using Lipofectamine 2000. After 4 days, the supernatant was collected, clarified

by centrifugation and filtered using a 0.22 μm filter. The resulting supernatant was added to Vero E6 cells grown overnight to 90–95% confluency in a 6-well plate. Cytosine arabinoside was added to the viral growth medium (DMEM, 5% FBS and 1% PS) at a concentration of 25 μg ml$^{-1}$ to inhibit growth of residual vaccinia virus. After 72 h, the wells were examined for evidence of VSV-MARV/Musoke GP replication by screening for GFP expression and viral cytopathic effect (CPE). The supernatant from a well showing GFP expression and viral CPE was collected, clarified by centrifugation, filtered using a 0.22 μm filter and stored at −80 °C until use. To increase the infectious titre of the rescued virus, the virus was serially passaged on Vero E6 cells as follows. Four million Vero E6 cells were split into a 10 cm plate and grown overnight at 37 °C and 5% $CO_2$ until they reached approximately 90–95% confluency. The rescued virus was added to the cells and incubated at 37 °C and 5% $CO_2$ for 2 h, after which the virus was removed and replaced with fresh viral growth medium. The cells were incubated for 72 h at 37 °C and 5% $CO_2$, after which the cells were examined for viral CPE and the supernatant was collected, clarified by centrifugation and filtered using a 0.45 μm filter. One-tenth of the resulting supernatant was added to new Vero E6 cells. The virus was passaged 5 times until significant viral CPE was observed, which indicated that the virus was efficiently replicating in the Vero E6 cells.

To isolate individual VSV-MARV/Musoke GP clones, the passaged virus was added to Vero E6 cells split into 6-well plates and grown overnight to 90–95% confluency. After 2 h, the virus was removed and an agarose overlay consisting of MEM, 5% FBS, 1% PS and 1% SeaPlaque agarose was added to the cells. The cells were incubated for 72 h, after which individual plaques were selected and the resulting virus was expanded on Vero E6 cells grown in 12-well plates to a confluency of 90–95%. Two individual clones, 2B2 and 2B4, were then further passaged twice on Vero E6 cells grown to 90–95% confluency in 15 cm plates as described above. After 2 passages, the resulting viral supernatant was collected, clarified by centrifugation, filtered using an 0.45 μm filter, concentrated 10-fold using a 100 kDa Amicon filter and stored at −80 °C until use. Infectious titre for the viral stocks were determined using plaque assays as follows. Vero E6 cells were split into 6-well plates and grown overnight to a confluency of 90–95%. The passaged virus was serially diluted in DMEM and added to the Vero E6 cells. The cells were incubated at 37 °C and 5% $CO_2$ for 2 h, after which the virus was removed and replaced with an agarose overlay. The cells were incubated for 72 h at 37 °C and 5% $CO_2$. Next, the cells were fixed with 2% paraformaldehyde for 15 min at room temperature and the agarose overlay was then removed. Cells were stained with an 0.1% crystal violet solution and subsequently washed 3 times with PBS, after which plaques were manually counted.

To select for escape mutations, Vero E6 cells were split into 12-well plates and grown overnight at 37 °C and 5% $CO_2$ until they reach 90–95% confluency. MARV16 alone, MARV16 and MR78, or MARV16 and MR191 were diluted to a starting concentration of 100 μg ml$^{-1}$ total antibody in DMEM and serially diluted 1:5 to a final concentration of 1.28 ng μl$^{-1}$. Approximately 10$^6$ PFU of VSV-MARV/Musoke GP isolate 2B2 or 2B4 were added to the serial diluted antibodies and the virus–antibody mixtures were incubated for 1 h at 37 °C. The virus–antibody mixtures were then added to the Vero E6 cells and incubated for 72 h at 37 °C and 5% $CO_2$. The cells were examined for CPE and the supernatant from the well with the highest antibody concentration that showed >20% CPE was collected, clarified by centrifugation and filtered using a 0.45 μm filter. The process was repeated using the collected viral supernatant until the well containing 100 μg ml$^{-1}$ of antibody showed >20% CPE, after which the supernatant was collected, clarified by centrifugation, filtered using a 0.45 μm filter and stored at −80 °C until use.

To identify escape mutations, viral RNA was extracted from the stored viral supernatant using a Zymo Quick-Viral RNA kit. cDNA was generated as previously described[82,83] using Turbo DNase to remove gDNA, Superscript IV with random hexamers to generate single-stranded

cDNA, Sequenase 2.0 polymerase to generate double-stranded cDNA and AMPure XP beads to purify the resulting cDNA. Sequencing libraries were generated using an Illumina DNA prep kit and sequenced on a 2 × 150 bp run on an Element Aviti. Sequencing reads were adapter-trimmed and quality-trimmed using Trimmomatic (v.0.39)[84] and mapped to the VSV-MARV/Musoke GP genome using Geneious Prime[85]. Variants present at a frequency of 10% or greater were identified using Geneious Prime.

## Reporting summary

Further information on research design is available in the Nature Portfolio Reporting Summary linked to this article.

## Data availability

The cryo-EM maps and atomic coordinates were deposited to the Electron Microscopy Data Bank and the PDB with accession numbers EMD-49486 and 9NJL, respectively. Sequencing reads are available under NCBI BioProject PRJNA1336301. Other data and materials generated in this study are available on request and may require a materials transfer agreement.

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

**Acknowledgements** This study was supported by the National Institute of Allergy and Infectious Diseases (DP1AI158186 and 75N93022C00036 to D.V.), an Investigators in the Pathogenesis of Infectious Disease Awards from the Burroughs Wellcome Fund (D.V.) and the University of Washington Arnold and Mabel Beckman cryo-EM center. D.V. is an Investigator of the Howard Hughes Medical Institute and the Hans Neurath Endowed Chair in Biochemistry at the University of Washington. Sequencing data reported in this publication were obtained from the Northwest Genomics Center at the University of Washington.

**Author contributions** A.A., L.P., D.C., F.B. and D.V. conceived the project. A.A., J.T.B. and C.S. recombinantly expressed glycoproteins. A.A. designed MARV GP constructs and evaluated their antigenicity and stability. B.C. and S.K. performed the BALB/c mice immunogenicity study. L.P. and A.D. performed ATX-GX mice immunization, monoclonal antibody isolation and ELISAs. K.C. and A.B. produced the recombinant antibodies. M.G. and R.C.J. carried out authentic virus neutralization assays. M.G., K.A., Y.G.-G. and R.C.J. performed the challenge study. A.A. conducted BLI binding experiments. A.A. and K.S. conducted the pseudovirus neutralization assays. B.P. performed the FcγRIIIa and FcγRIIa activation assays. A.A. conducted the antibody escape experiments. A.A. carried out negative-stain and cryo-EM sample preparation, data collection and processing with help from Y.-J.P. and M.M. A.A. and D.V. built and refined the structures. A.A. and D.V. wrote the manuscript with input from all authors. A.A., L.P., D.C., F.B. and D.V. analysed the data.

**Competing interests** L.P., B.P., A.D., K.C., A.B., D.C. and F.B. are employees of Vir Biotechnology and may hold shares. Vir Biotechnology (L.P., D.C. and F.B.) and the University of Washington (A.A. and D.V.) filed a provisional patent application describing these results.

### Additional information
**Correspondence and requests for materials** should be addressed to Fabio Benigni or David Veesler.

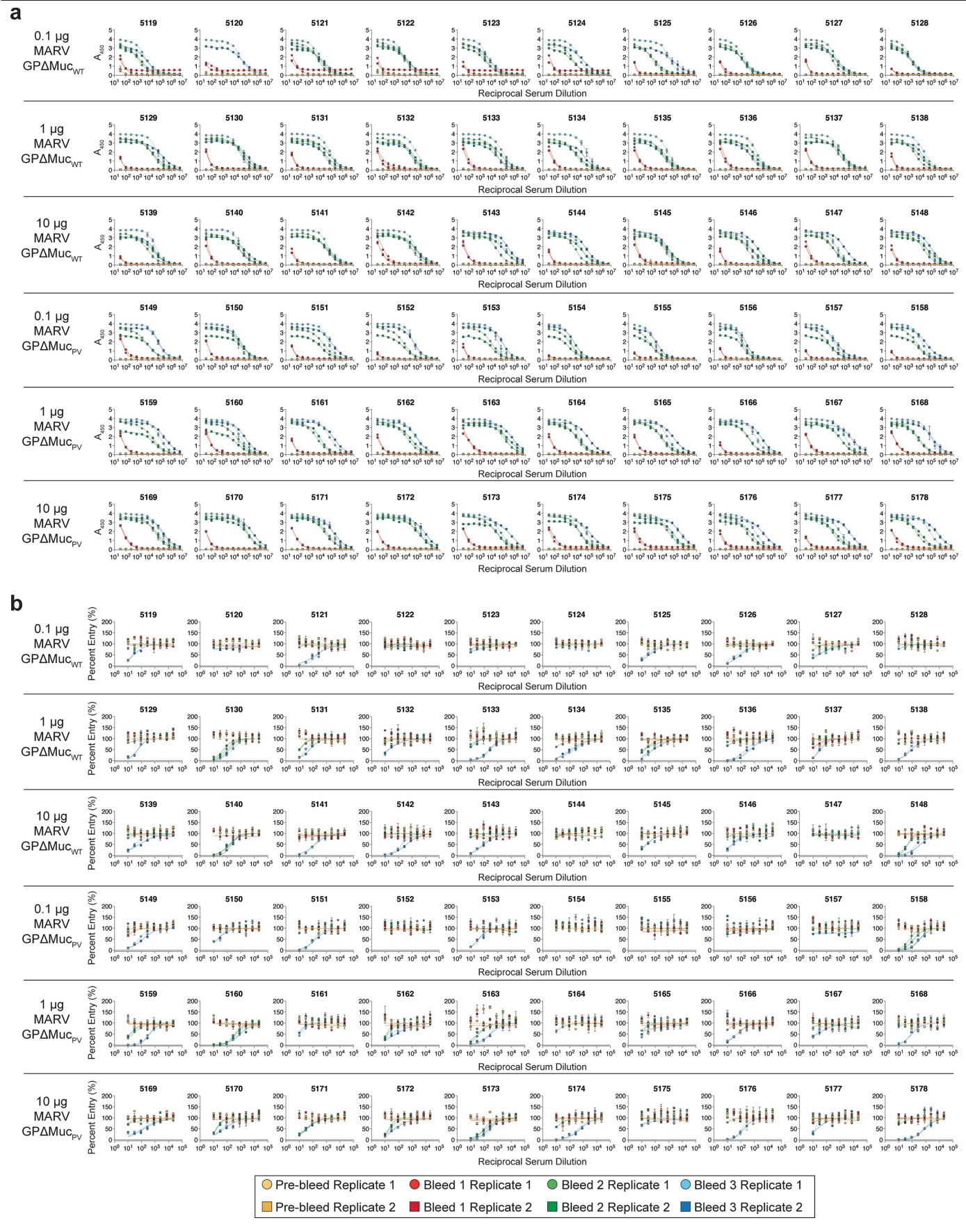

**Extended Data Fig. 1** | See next page for caption.

**Extended Data Fig. 1 | Analysis of serum binding and neutralization titres for MARV GPΔMuc$_{WT}$- and GPΔMuc$_{PV}$-immunized mice. a**,**b**, Dose-response curves for the ELISAs against the MARV GPΔMuc$_{PV}$ ectodomain (**a**) and for the neutralization assays against VSV pseudotyped with MARV/Musoke GP (**b**) for the sera collected from MARV GPΔMuc$_{WT}$- or GPΔMuc$_{PV}$-immunized mice. Serum was unable to be collected at bleed 2 for mouse D5120. Two biological replicates were performed for the ELISAs using distinct batches of proteins. Two technical replicates were performed per biological replicate. Data presented are from one representative biological replicate and presented as mean ± standard error from the two technical replicates. Two to six biological replicates were performed for the neutralization assays using distinct batches of antibodies and pseudoviruses. Three technical replicates were performed per biological replicate. Data from all biological replicates are shown and presented as mean ± standard error from the three technical replicates.

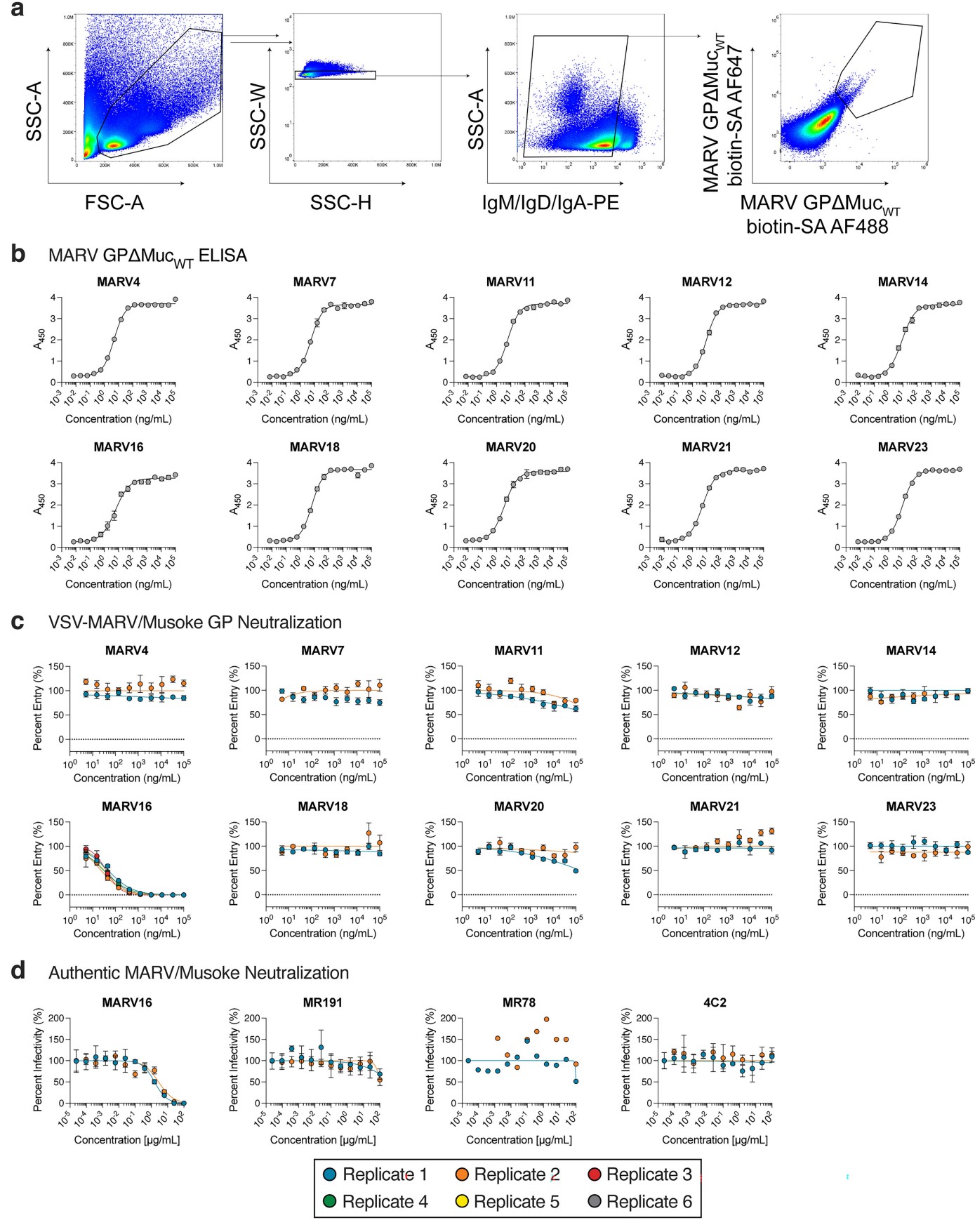

**a**

**b** MARV GPΔMuc_WT ELISA

**c** VSV-MARV/Musoke GP Neutralization

**d** Authentic MARV/Musoke Neutralization

**Extended Data Fig. 2** | See next page for caption.

**Extended Data Fig. 2 | Analysis of binding and neutralization titres for the monoclonal antibodies discovered from ATX-Gx mice. a**, Flow cytometry gating strategy used for sorting MARV GPΔMuc$_{WT}$-reactive memory B cells. **b,c**, Dose-response curves for the ELISAs against the MARV GPΔMuc$_{WT}$ ectodomain (**b**) and neutralization assays against VSV pseudotyped with MARV/Musoke GP (**c**) for the 10 antibodies discovered from the immunization study using the ATX-Gx mice. Two biological replicates were performed for the ELISAs using distinct batches of proteins and antibodies. Two technical replicates were performed per biological replicate. Data presented are from one representative biological replicate and presented as mean ± standard error from the two technical replicates. Two to six biological replicates were performed for the neutralization assays using distinct batches of antibodies and pseudoviruses. Three technical replicates were performed per biological replicate. Data from all biological replicates are shown and presented as mean ± standard error from the three technical replicates. **d**, Dose-response curves for plaque reduction neutralization tests for MARV16, MR78, MR191, and the MERS-CoV 4C2 IgG, conducted using authentic MARV/Musoke. Two biological replicates were performed with one to three technical replicates using distinct batches of monoclonal antibodies. Data are shown as the mean ± standard error of the technical replicates.

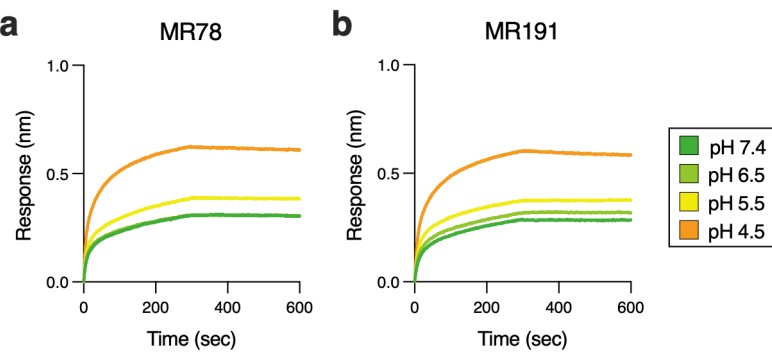

**Extended Data Fig. 3 | MR78 and MR191 binding to MARV GPΔMuc$_{WT}$ at different pHs. a,b**, Binding of MR78 (**a**) and MR191 (**b**) IgGs at a concentration of 100 nM at the indicated pH to immobilized MARV GPΔMuc$_{WT}$, as measured by biolayer interferometry. Data shown are one representative out of two biological replicates using distinct batches of protein and antibodies.

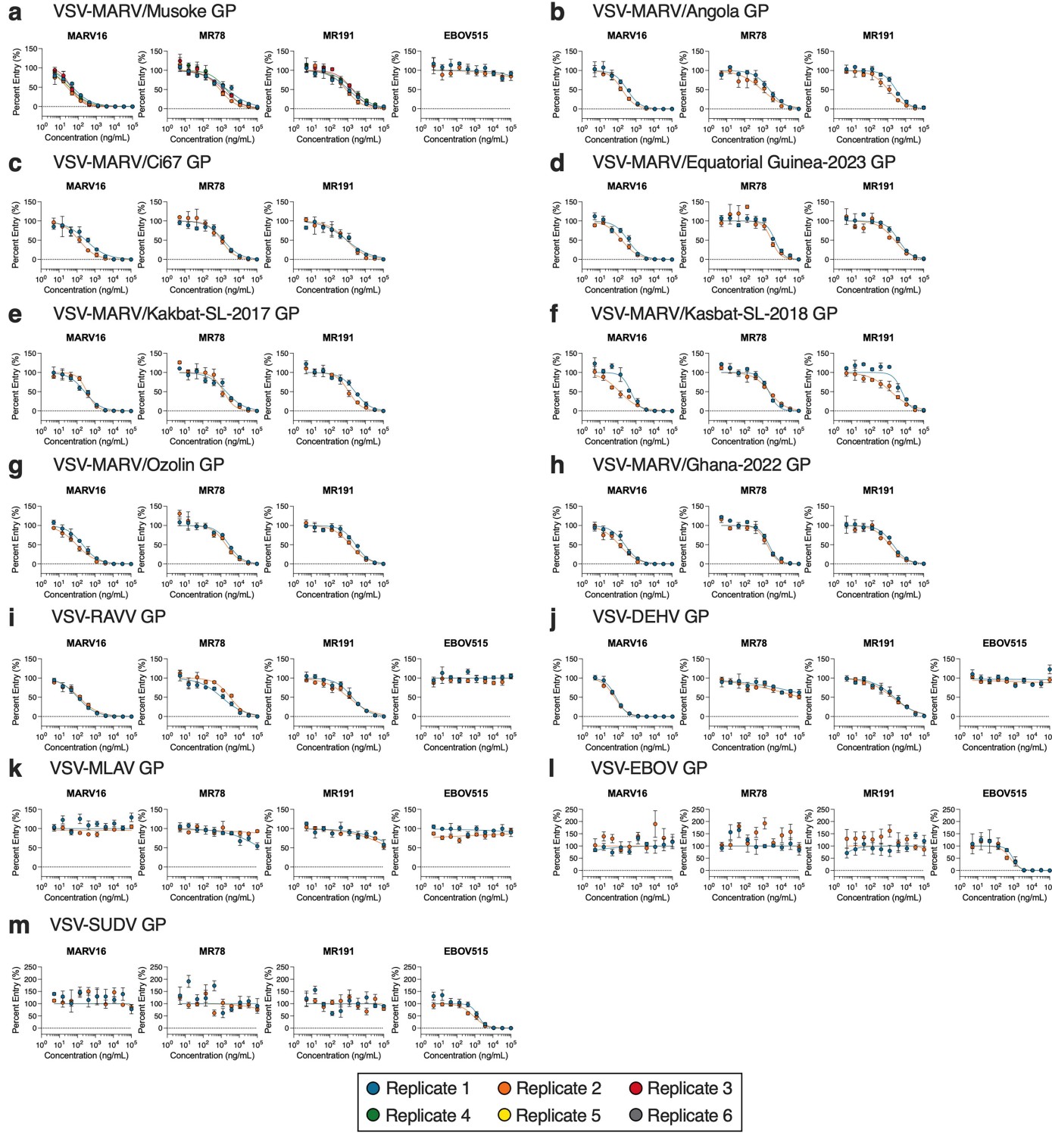

**Extended Data Fig. 4 | Analysis of neutralization breadth of MARV monoclonal antibodies. a–m**, Neutralization dose-response curves for MARV16, MR78, MR191, and EBOV515 against VSV pseudotyped with the MARV/Musoke GP (**a**), MARV/Angola GP (**b**), MARV/Ci67 GP (**c**), MARV/Equatorial Guinea-2023 GP (**d**), MARV/Kakbat-SL-2017 GP (**e**), MARV/Kasbat-SL-2018 GP (**f**), MARV/Ozolin GP (**g**), MARV/Ghana-2022 GP (**h**), RAVV GP (**i**), DEHV GP (**j**), MLAV GP (**k**), EBOV GP (**l**), or SUDV GP (**m**). Each of the two to six biological replicates used distinct batches of pseudoviruses and antibodies and data are shown as the mean ± standard error of technical triplicates.

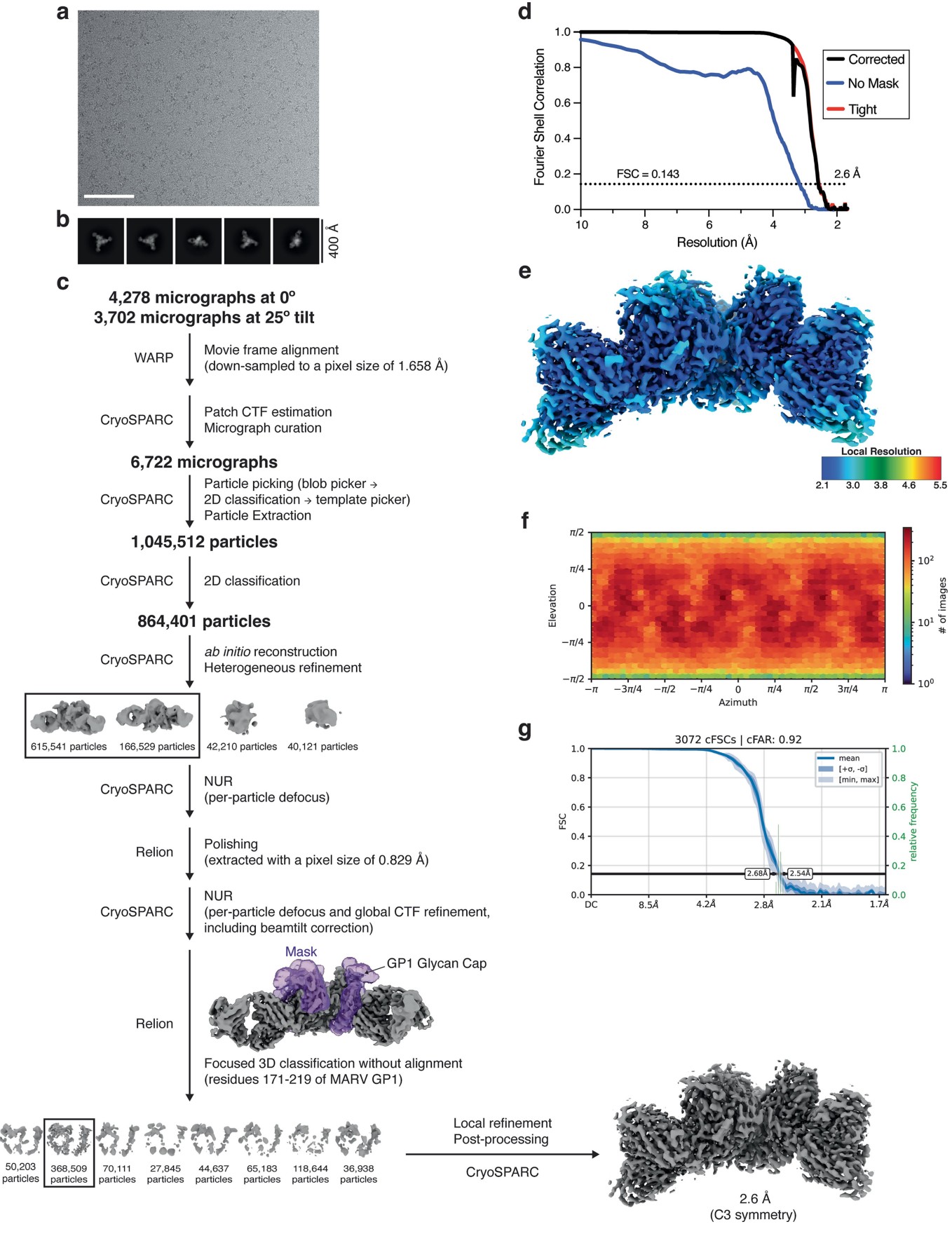

**Extended Data Fig. 5** | See next page for caption.

**Extended Data Fig. 5 | Cryo-EM data processing of the MARV GPΔMuc_WT-MARV16 Fab complex. a,b,** Representative cryo-EM micrograph (**a**) and 2D class averages (**b**) obtained for MARV GPΔMuc_WT in complex with MARV16 Fabs. Scale bar: 100 nm. **c,** Cryo-EM data processing workflow. NUR (per-particle defocus): non-uniform refinement with per-particle defocus refinement. **d,** Gold-standard fourier shell correlation curves for the locally refined MARV GPΔMuc_WT-MARV16 Fab complex (using a mask comprising the GP trimer and MARV16 Fab variable domains). **e,** Local resolution calculated with cryoSPARC for the locally refined MARV GPΔMuc_WT-MARV16 Fab complex. **f,** Heat map of angular distribution for the particles contributing to the final reconstruction. **g,** Conical fourier shell correlation plot[86].

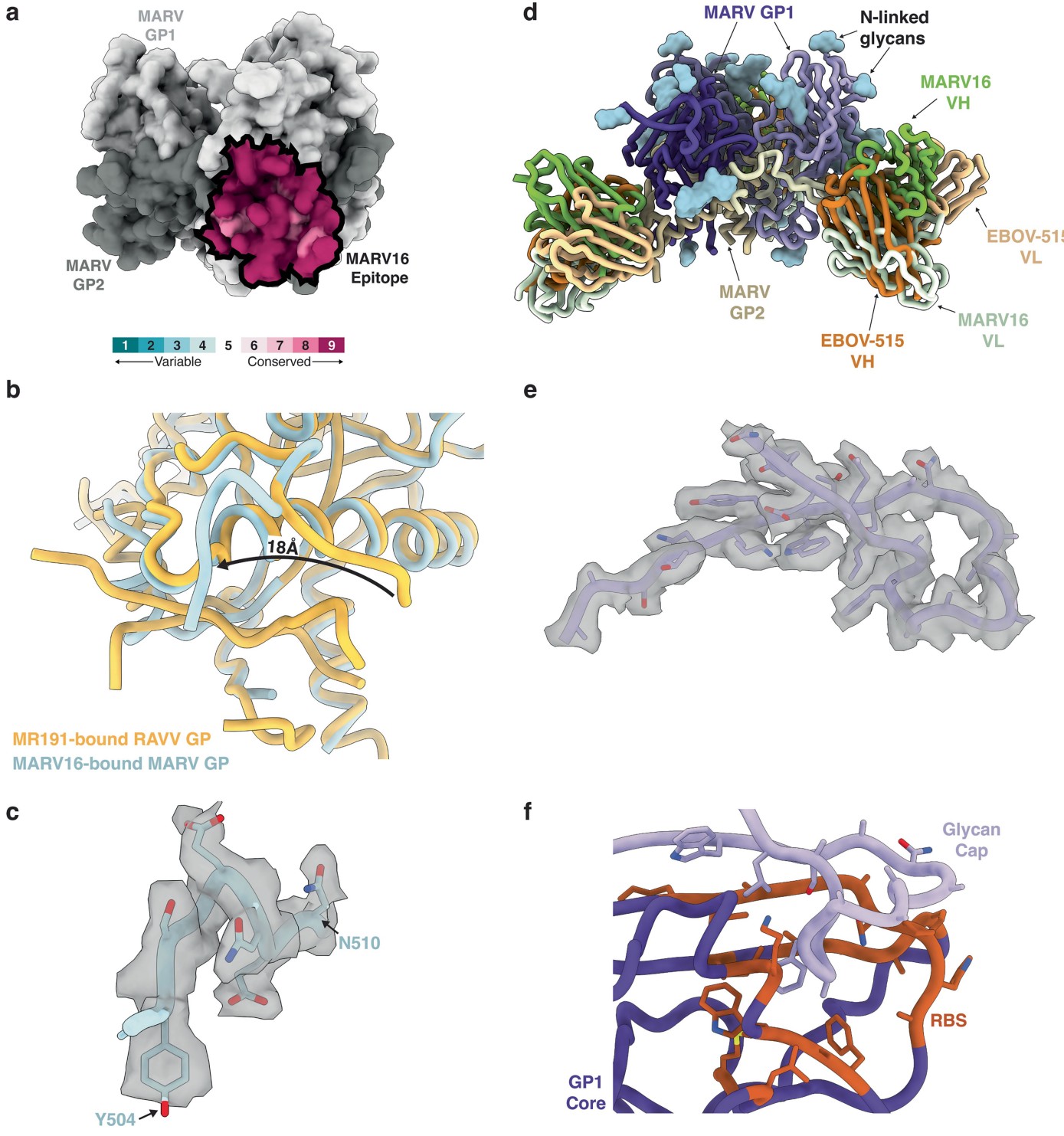

**a**, MARV GP1, MARV GP2, MARV16 Epitope

Variable 1 2 3 4 5 6 7 8 9 Conserved

**b**, MR191-bound RAVV GP, MARV16-bound MARV GP, 18Å

**c**, N510, Y504

**d**, MARV GP1, N-linked glycans, MARV16 VH, EBOV-515 VL, MARV GP2, MARV16 VL, EBOV-515 VH

**f**, Glycan Cap, GP1 Core, RBS

**Extended Data Fig. 6 | Analysis of the MARV16-bound MARV GPΔMuc_WT complex. a**, Sequence conservation of MARV GP residues comprising the MARV16 epitope across the *Marburgvirus* isolates (MARV variants and RAVV) assessed in this study. Conservation scores were assigned using ConSurf[87]. **b**, Superimposition of the MR191 Fab-bound RAVV GP (PBD: 6BP2; orange) and MARV16 Fab-bound MARV GP (blue) comparing the N-terminus of the GP2 core domain of the two models. The arrow indicates the shift of the equivalent residues in each model. The MR191 and MARV16 Fabs are not shown for clarity. **c**, MARV GP2 residues 504–510 modelled into the cryo-EM map (semi-transparent grey surface). **d**, Comparison of the binding modes of MARV16 (green) and the

EBOV antibody EBOV-515 (orange). The EBOV GP trimer from the EBOV GP-EBOV-442-EBOV-515 complex structure (PDB: 7M8L) was superimposed with the MARV GP trimer from the MARV GPΔMuc_WT-MARV16 structure to compare the EBOV-515 and MARV16 binding poses. MARV GP1 and GP2 are shown in different shades of purple and beige, respectively. N-linked glycans are rendered as light blue surfaces. **e**, The MARV GP1 glycan cap (residues 191–219) modelled into the cryo-EM density (semi-transparent grey surface). **f**, View of the putative RBS residues (shown in orange) that are shielded by the MARV glycan cap (light purple). The rest of GP1 is rendered purple.

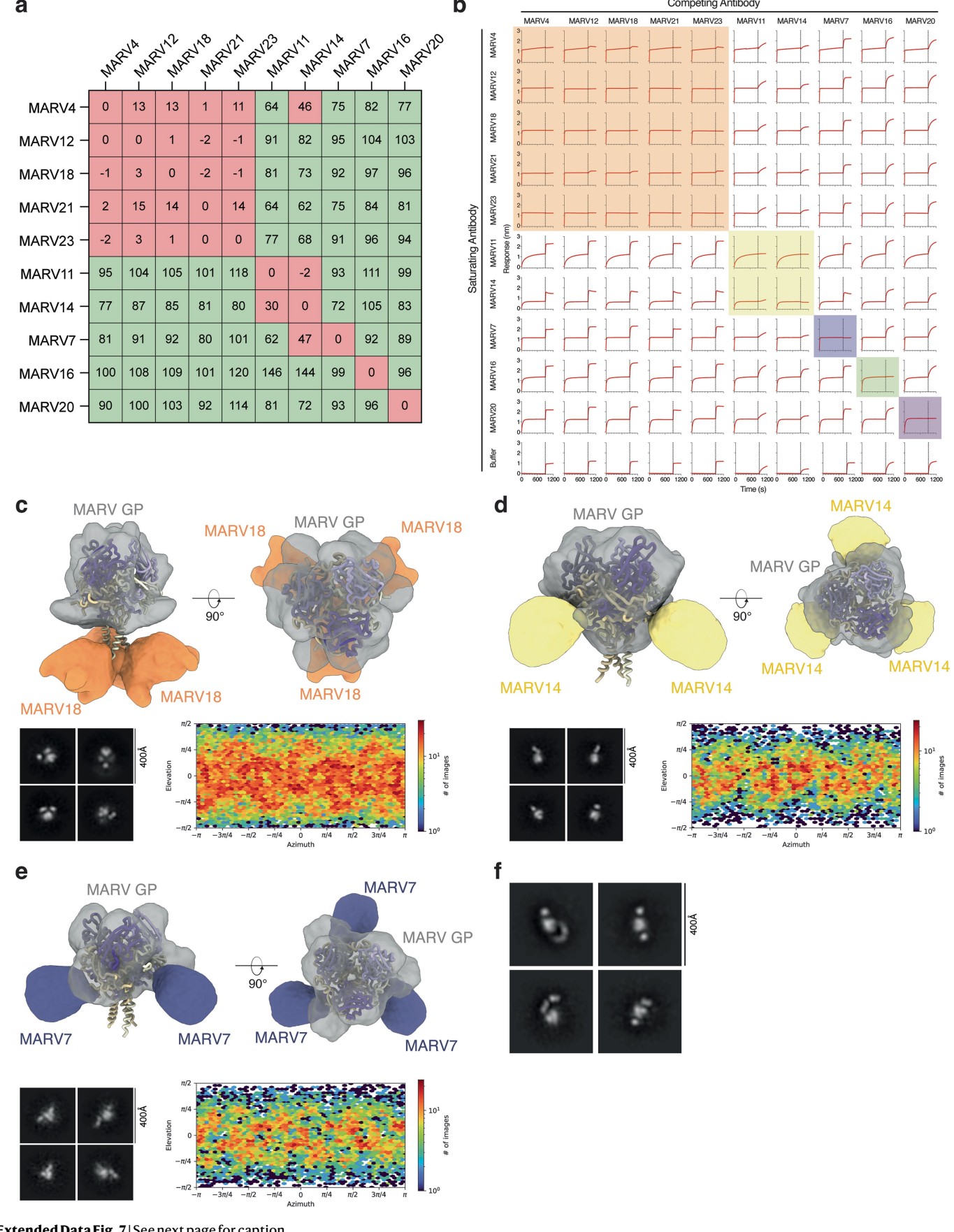

**Extended Data Fig. 7** | See next page for caption.

**Extended Data Fig. 7 | Identification of MARV GPΔMuc$_{WT}$ antigenic sites recognized by non-neutralizing monoclonal antibodies. a**,**b**, Percent binding (**a**) to the MARV GPΔMuc$_{WT}$ and BLI traces (**b**) for antibody pairs evaluated in the epitope binning experiments. Antibody pairs showing reciprocal blocking relationships were classified as belonging to the same binding group. Data from one biological replicate are shown and representative of two biological replicates. **c**–**e**, 3D reconstructions, representative 2D class averages, and angular distribution plots obtained by single particle electron microscopy analysis of negatively stained MARV GPΔMuc$_{WT}$ bound to the MARV18 (**c**), MARV14 (**d**), or MARV7 (**e**) Fabs. **f**, Representative 2D classes from an electron microscopy dataset of negatively-stained MARV GPΔMuc$_{WT}$-MARV20 Fab complex.

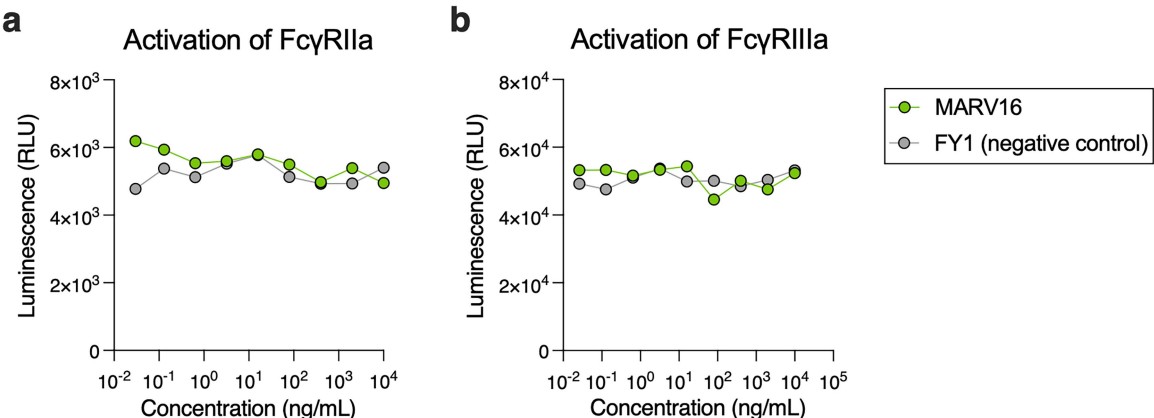

**Extended Data Fig. 8 | MARV16 does not activate FcγRIIa or FcγRIIIa.**
**a,b**, In vitro evaluation of MARV16 mAb-mediated activation of human FcγRIIIa
V158 (**a**) and FcγRIIa H131 (**b**) using a bioluminescent reporter assay as a surrogate assay for Fc-mediated effector functions. ExpiCHO cells transfected with
MARV/Musoke GP and Jurkat-Fcγ cells were used as target and effector cells,
respectively.

**Extended Data Table 1 | VH and VL amino acid sequences for the 10 MARV antibodies discovered in this study**

| | VH Sequence | VL Sequence |
|---|---|---|
| **MARV4** | QVQLVESGGGVVQPGRSLRLSCAASGFTFTTYGIHWVRQAPGKGLEWVAVISFDGNNKYYADSVKGRFTISRDNSKNTLYLQMNSLRAEDSAVYYCAKDQGSRSPGVFDYWGQGTLVTVSS | DIVMTQSPDSLAVSLGERATINCKSSQSVLYSSNNKNFLTWYQQKPRQPPKLLINWASTRESGVPDRFSGSGSGTDFTLTISSLQAEDVAVYYCQQFYNAPWTFGQGTKVDIK |
| **MARV7** | EVQLVESGGGLVKPGGSLRLSCAASGFTFSNAWMSWVRQAPGKGLEWVGRIKRKTDGGTTDYAAPVKGRFTISRDDSKNTLYLQMNSLKTEDTAVYYCTTTGTMWGQGTMVTVSS | DIVMTQTPLSLPVTLGQPASISCRSSQSLVYSDGNTYLNWFQQRPGQSPRRLIYKVSNRDSGVPDRFSGSGSGTDFTLKISRVEAEDVGVYYCMQGTHWPPYTFGQGTKLEIK |
| **MARV11** | EVQLVQSGGGLVQPGGSLRLSCAASGFTFSRFDMHWVRQAPGKGLEWVSSIGTAGETFYSGSVKGRFTISRENAKNSFYLQMNSLRAGDTAEYYCSRGLDYWGQGTLVTVSS | DIVMTQSPLSSPVTLGQPASISCRSSQSLLHSDGNTYLNWLQQRPGQPPGLLIYKISNRFSGVPDRFSGSGAGTDFTLKISRVEAEDVGVYYCMQATQFLITFGQGTKVDIK |
| **MARV12** | QVQLVESGGGLVQPGGSLRLSCAASGFTFSNAFMSWVRQAPGKGLEWVGRIKSNTYGGTSVYAAPVKGRFTISRDDSTNTLYLQMNGLKTEDTAVYYCEGTSLYFDYWGQGTLVTVSS | DIVMTQSPLSSLVTLGQPASISCRSSQSLVHSDGNTYLSWLQQRPGQPLRLLIYKISNRFSGVPDRFSGSGAGTDFTLKISRVEAEDVGVYYCMQTTQFPTFGGGTKVDIK |
| **MARV14** | EVQLVESGGGLVQPGGSLRLSCAASGFTFSRYDMHWVRQATGKGLEWVSAIGTAGDTYYPGSVKGRFTISRENAKNSLYLQMNSLRAGDTAVYYCSRGLDYWGQGTLVTVSS | DIVMTQSPLSSPVTLGQPASISCRSSQSLVHSDGTTYLNWLQQRPGQPPRLLIYKISNRFSGVPDRFSGSGAGTDFTLKISRVEAEDVGVYYCMQATQFPVTFGPGTKVDIK |
| **MARV16** | EVQLVESGGGLVKPGGSLRLSCAASGFTFSSYTMNWVRQAPGKGLEWVSSISSSESYIYYVDSVQGRFTISRDNAKNSLYLQLNSLRAEDTAVYYCVRRDNWNYDTLDIWGQGTMVTVSS | DIVMTQSPSSLSASVGDRVIITCRASQSISTYLNWYQQKPGKAPKLLIYAASSLQSGVPSRFSGSESGTDFTLTISGLQPEDFATYYCLQSYDAPLTFGGGTKVDIK |
| **MARV18** | EVQLVESGGAVVQPGRSLRLSCAASGFSFSTYGMHWVRQAPGKGLEWVALISYDGTNKFYADSVKGRFTISRDNSNNTLFLQMNSLRPEDTALYYCAKDGGFDYYGGWLDPWGQGTLVTVSS | DIVMTQSPDSLAVSLGERATINCKSSQNILYSSNNKNYLTWYQQKPGQPPKLLIHWASTRESGVPDRFSGSGSGTDFTLTISSLQAEDVAVYYCQQYYRSPLTFGGGTKVDIK |
| **MARV20** | EVQLVESGGGLVQPGGSLKLSCAASGFTFSGSALHWVRQVSGKGLEWVGRLRSKTNNYATAYAASVKGRFTISRDDSKNTAYLQMNSLKSEDTAVYYCTSPSIKPAGTDFDYWGQGALVTVSS | DIVMTQSPSSLSASVGDRVTITCRASQGISHYLAWFQQKPGKAPKSLIFAASSLQSGVPSKFSGSGSGTDFTLTISSLQPEDFATYYCQQYNSDPLTFGGGTKVDIK |
| **MARV21** | QVQLVQSGGGLVQPGGSLRLSCAASGFTFSSYWMSWVRQAPGKGLEWVANIKQDGSEKYYVDSVKGRFTISRDNAKNSLYLQMNSLRAEDTAVYYCASNWGGDYWGQGTLVTVSS | DIVMTQTPDSLAVSLGERATINCKSSQSVLYSSNNKNYLAWYQQKPGQPPKLLIYWASTRESGVPDRFSGSGSGTDFTLTISSLQAEDVAVYYCQQYYSTPYTFGQGTKLEIK |
| **MARV23** | EVQLVESGGGVVQPGRSLRLACVASGFTFSRYGIHWVRQAPGKGLEWVAVISYDGSNKYYTDSVKGRFTISRDNSKNTVFLQMNSLRTEDTAVYYCVKDQGSRSPGIFDYWGQGTPVTVSS | DIVMTQSPDSLAVSLGERATINCKSSQSLLYSSNNKNYLTWFQQKPGQPPKLLIYWASTRESGVPDRFSGSGSGTDFTLTIGSLQAEDVAVYYCQQCYHAPYTFGQGTKLEIK |

**Extended Data Table 2 | Binding kinetics of the MARV16 Fab to immobilized MARV GPΔMuc$_{WT}$ determined by BLI**

| K$_D$ (nM) | k$_{on}$ (M$^{-1}$s$^{-1}$) | k$_{off}$ (s$^{-1}$) |
|---|---|---|
| 1.35 ± 0.10 | 1.27 x 10$^5$ | 1.70 x 10$^{-4}$ |

Values are presented as mean±standard deviation obtained from two biological replicates using distinct batches of protein.

**Extended Data Table 3 | Cryo-EM data collection, refinement and validation statistics**

| | MARV GPΔMuc$_{WT}$-MARV16 (EMDB-49486) (PDB 9NJL) |
|---|---|
| **Data collection and processing** | |
| Magnification | 105,000 |
| Voltage (kV) | 300 |
| Electron exposure (e–/Å$^2$) | 63 |
| Defocus range (μm) | -0.4 to -3.0 |
| Pixel size (Å) | 0.829 |
| Symmetry imposed | *C3* |
| Initial particle images (no.) | 1,045,512 |
| Final particle images (no.) | 368,509 |
| Map resolution (Å) | 2.6 |
| FSC threshold | 0.143 |
| | |
| **Refinement** | |
| Initial model used (PDB code) | 6BP2 |
| Model resolution (Å) | 2.7 |
| FSC threshold | 0.5 |
| Map sharpening *B* factor (Å$^2$) | -82.9 |
| Model composition | |
| Non-hydrogen atoms | 11,307 |
| Protein residues | 1,455 |
| Glycans | 36 |
| *B* factors (Å$^2$) | |
| Protein | 26.45 |
| Ligand | 76.9 |
| R.m.s. deviations | |
| Bond lengths (Å) | 0.005 |
| Bond angles (°) | 1.063 |
| Validation | |
| MolProbity score | 2.13 |
| Clashscore | 9.04 |
| Poor rotamers (%) | 2.65 |
| Ramachandran plot | |
| Favored (%) | 95.35 |
| Allowed (%) | 4.65 |
| Disallowed (%) | 0 |

**Extended Data Table 4 | Residue contact table**

| MARV GP | | MARV16 | |
| --- | --- | --- | --- |
| **Residue** | **Subunit** | **Residue** | **Chain** |
| S55 | GP1 | G26 | Heavy |
| G56 | GP1 | F27 | Heavy |
| K58 | GP1 | T28 | Heavy |
| E87 | GP1 | S30 | Heavy |
| E88 | GP1 | S31 | Heavy |
| K90 | GP1 | Y32 | Heavy |
| K120 | GP1 | T33 | Heavy |
| S180 | GP1 | S52 | Heavy |
| Y504 | GP2 | S53 | Heavy |
| S505 | GP2 | S54 | Heavy |
| G506 | GP2 | E55 | Heavy |
| E507 | GP2 | S56 | Heavy |
| E509 | GP2 | Y57 | Heavy |
| N510 | GP2 | Y59 | Heavy |
| C512 | GP2 | N74 | Heavy |
| D513 | GP2 | R98 | Heavy |
| A514 | GP2 | R99 | Heavy |
| E515 | GP2 | N101 | Heavy |
| L516 | GP2 | W102 | Heavy |
| R517 | GP2 | Y104 | Heavy |
| L548 | GP2 | D105 | Heavy |
| K550 | GP2 | Y32 | Light |
| N551 | GP2 | S91 | Light |
| Q552 | GP2 | Y92 | Light |
| N554 | GP2 | D93 | Light |
| C557 | GP2 | | |
| R560 | GP2 | | |

# Reporting Summary

## Statistics

For all statistical analyses, confirm that the following items are present in the figure legend, table legend, main text, or Methods section.

| n/a | Confirmed | |
|---|---|---|
| ☐ | ☒ | The exact sample size (*n*) for each experimental group/condition, given as a discrete number and unit of measurement |
| ☐ | ☒ | A statement on whether measurements were taken from distinct samples or whether the same sample was measured repeatedly |
| ☐ | ☒ | The statistical test(s) used AND whether they are one- or two-sided<br>*Only common tests should be described solely by name; describe more complex techniques in the Methods section.* |
| ☐ | ☒ | A description of all covariates tested |
| ☐ | ☒ | A description of any assumptions or corrections, such as tests of normality and adjustment for multiple comparisons |
| ☐ | ☒ | A full description of the statistical parameters including central tendency (e.g. means) or other basic estimates (e.g. regression coefficient) AND variation (e.g. standard deviation) or associated estimates of uncertainty (e.g. confidence intervals) |
| ☐ | ☒ | For null hypothesis testing, the test statistic (e.g. *F*, *t*, *r*) with confidence intervals, effect sizes, degrees of freedom and *P* value noted<br>*Give P values as exact values whenever suitable.* |
| ☒ | ☐ | For Bayesian analysis, information on the choice of priors and Markov chain Monte Carlo settings |
| ☒ | ☐ | For hierarchical and complex designs, identification of the appropriate level for tests and full reporting of outcomes |
| ☒ | ☐ | Estimates of effect sizes (e.g. Cohen's *d*, Pearson's *r*), indicating how they were calculated |

*Our web collection on statistics for biologists contains articles on many of the points above.*

## Software and code

Policy information about availability of computer code

| | |
|---|---|
| Data collection | Leginon; QuantStudio Design and Analysis Desktop Software; FACS DIVA (Version 9.0) |
| Data analysis | GraphPad Prism 10; Octet Data Analysis HT software v12.0; WARP; cryoSPARC; Relion; UCSF ChimeraX; Coot; AlphaFold2; Rosetta; ISOLDE; Phenix; QuantStudio Design and Analysis Desktop Software; Trimmomatic v0.39; Geneious Prime; FlowJo (version 10.8.0) |

For manuscripts utilizing custom algorithms or software that are central to the research but not yet described in published literature, software must be made available to editors and reviewers. We strongly encourage code deposition in a community repository (e.g. GitHub). See the Nature Portfolio guidelines for submitting code & software for further information.

## Data

Policy information about availability of data

All manuscripts must include a data availability statement. This statement should provide the following information, where applicable:
- Accession codes, unique identifiers, or web links for publicly available datasets
- A description of any restrictions on data availability
- For clinical datasets or third party data, please ensure that the statement adheres to our policy

The cryo-EM maps and atomic coordinates were deposited to the Electron Microscopy Data Bank (EMDB) and the PDB with accession numbers 9NJL and EMD-49486. Sequencing reads are available under NCBI BioProject PRJNA1336301.

# Research involving human participants, their data, or biological material

Policy information about studies with human participants or human data. See also policy information about sex, gender (identity/presentation), and sexual orientation and race, ethnicity and racism.

| | |
|---|---|
| Reporting on sex and gender | N/A |
| Reporting on race, ethnicity, or other socially relevant groupings | N/A |
| Population characteristics | N/A |
| Recruitment | N/A |
| Ethics oversight | N/A |

Note that full information on the approval of the study protocol must also be provided in the manuscript.

# Field-specific reporting

Please select the one below that is the best fit for your research. If you are not sure, read the appropriate sections before making your selection.

☒ Life sciences  ☐ Behavioural & social sciences  ☐ Ecological, evolutionary & environmental sciences

For a reference copy of the document with all sections, see nature.com/documents/nr-reporting-summary-flat.pdf

# Life sciences study design

All studies must disclose on these points even when the disclosure is negative.

| | |
|---|---|
| Sample size | No sample size calculation was performed to design the study. The number of mice used for antibody discovery and immunogenicity study and the number of Guinea pigs used for the challenge study were based on previous experience with the model. |
| Data exclusions | No data were excluded. |
| Replication | Experimental assays were performed at least in two independent biological replicates. Each replicate was performed with 1 to 6 technical replicates. All attempts at replication were successful. |
| Randomization | For the immunogenicity study, mice were randomly assigned to groups ensuring the groups had an equal number of females and males. For the challenge study, Guinea pigs were randomly assigned to groups. For the antibody discovery study, no randomization was performed as all antigen-positive memory B cells were evaluated. |
| Blinding | Blinding was not performed as this study is not a case control study. |

# Reporting for specific materials, systems and methods

We require information from authors about some types of materials, experimental systems and methods used in many studies. Here, indicate whether each material, system or method listed is relevant to your study. If you are not sure if a list item applies to your research, read the appropriate section before selecting a response.

## Materials & experimental systems

| n/a | Involved in the study |
|---|---|
| ☐ | ☒ Antibodies |
| ☐ | ☒ Eukaryotic cell lines |
| ☒ | ☐ Palaeontology and archaeology |
| ☐ | ☒ Animals and other organisms |
| ☒ | ☐ Clinical data |
| ☒ | ☐ Dual use research of concern |
| ☒ | ☐ Plants |

## Methods

| n/a | Involved in the study |
|---|---|
| ☒ | ☐ ChIP-seq |
| ☐ | ☒ Flow cytometry |
| ☒ | ☐ MRI-based neuroimaging |

# Antibodies

| | |
|---|---|
| Antibodies used | MARV4 (generated in-house); MARV7 (generated in-house); MARV11 (generated in-house); MARV12 (generated in-house); MARV14 |

| Antibodies used | (generated in-house); MARV16 (generated in-house); MARV18 (generated in-house); MARV20 (generated in-house); MARV21 (generated in-house); MARV23 (generated in-house); MR78 (PDB: 5UQ, generated in-house); MR191 (PDB: 6BP2, generated in-house); EBOV-515 (PBD: 7KF9, generated in-house); Goat anti-human IgG Fc HRP (Thermo Fisher Scientific, catalog #A18823, 1:5,000 dilution);PE anti-mouse IgM Antibody (BioLegend UK LTD, catalog #406508); IgA Monoclonal Antibody (mA-6E1), PE, eBioscience™ (Fisher Scientific AG, catalog #12-4204-83); PE anti-mouse IgD Antibody (BioLegend UK LTD, catalog #405706); APC anti-mouse CD19 Antibody (BioLegend UK LTD, catalog #152410); PE anti-mouse Ig light chain λ Antibody (BioLegend UK LTD, catalog #407308); 4C2 (PDB: 5DO2; generated in-house); FY1 (generated in-house);  MGH2 (generated in-house); Goat anti-Mouse IgG (H+L) Poly-HRP Secondary Antibody (ThermoFisher; catalog #32230) |
|---|---|
| Validation | MARV4, MARV7, MARV11, MARV12, MARV14, MARV16, MARV18, MARV20, MARV21, MARV23, MR78, MR191, EBOV-515, 4C2, FY1, and MGH2 binding to the target antigen was validated with multiple binding assays. The binding epitope of MARV16 was further identified cryoEM.<br><br>The reactivity of the remaining antibodies were validated by the manufacturer. Validation information is available on the manufacturer's website. |

## Eukaryotic cell lines

Policy information about cell lines and Sex and Gender in Research

| Cell line source(s) | HEK-293T, VeroE6, and BS-C-1 cells were obtained from ATCC. Expi293 cells were obtained from Thermo Fisher Scientific. BHK-21/WI-2 cells were obtained from Kerafast. |
|---|---|
| Authentication | None of the cell line used were authenticated. |
| Mycoplasma contamination | Cells lines were not tested for mycoplasma contamination. |
| Commonly misidentified lines (See ICLAC register) | No commonly misidentified cells lines were used in this study. |

## Animals and other research organisms

Policy information about studies involving animals; ARRIVE guidelines recommended for reporting animal research, and Sex and Gender in Research

| Laboratory animals | For the antibody discovery study, ATX-GK and ATX-GL female mice, 6-7 weeks old, were obtained from Alloy Therapeutics Inc.<br><br>For the immunogenicity study, 6-8 -week-old BALB/C mice (Mus musculus) were obtained from Inotiv.<br><br>For the challenge study, 4-8 week old Hartley Guinea Pigs (Cavia porcellus) were obtained from Charles River Laboratories. |
|---|---|
| Wild animals | This study did not include wild animals. |
| Reporting on sex | Only female mice were used for the antibody discovery study.<br><br>For the immunogenicity study, 30 male and 30 female mice were used. Each group contained 5 male and 5 female mice.<br><br>Only female Hartley Guinea Pigs were used for the challenge study. |
| Field-collected samples | This study did not involve field-collected samples. |
| Ethics oversight | For the antibody discovery work, animal experiments were performed in accordance with the Swiss Federal Veterinary Office guidelines and authorized by the Cantonal Veterinary (approval no. 35554 TI-39/2023/2023).<br><br>For the immunogenicity study, experiments were performed under the BIOQUAL Institutional Animal Care and Use Committee (IACUC)-approved Protocol no. 23-054P.<br><br>For the challenge study, experiments were approved by Texas Biomed's IACUC under protocol #1915C prior to the initiation of the study and performed in accordance with the Animal Welfare Act and the Guide for the Care and Use of Laboratory Animals. |

Note that full information on the approval of the study protocol must also be provided in the manuscript.

## Plants

| | |
|---|---|
| Seed stocks | N/A |
| Novel plant genotypes | N/A |
| Authentication | N/A |

## Flow Cytometry

### Plots

Confirm that:

☒ The axis labels state the marker and fluorochrome used (e.g. CD4-FITC).

☒ The axis scales are clearly visible. Include numbers along axes only for bottom left plot of group (a 'group' is an analysis of identical markers).

☒ All plots are contour plots with outliers or pseudocolor plots.

☒ A numerical value for number of cells or percentage (with statistics) is provided.

### Methodology

| | |
|---|---|
| Sample preparation | As described in the method section, the mice were sacrificed and peripheral blood, spleen and lymph nodes (LN) were collected and cells freshly isolated. B cells from either freshly isolated or frozen splenocytes were enriched by positive selection using mouse CD19 microbeads and LS columns (Miltenyi) and subsequently stained with mouse anti-IgM, anti-IgD, anti-IgA and biotinylated MARV GPΔMuc labeled with both streptavidin- Alexa-Fluor 488 and streptavidin-Alexa-Fluor 647 (Life Technologies). MARV GPΔMuc -specific IgG+ memory B cells were sorted by flow cytometry via gating out IgM/IgD/IgA-positive B cells and positively baiting B cells with dual-labeled (Alexa-Fluor 488 and Alexa-Fluor 647) antigen, using SH800SFP cell sorter (Sony). |
| Instrument | Samples were acquired on FACS Symphony A1 (BD) |
| Software | Flow cytometry data was generated using FACS DIVA (Version 9.0). The flow cytometry data were analysed using FlowJo (version 10.8.0). |
| Cell population abundance | From Spleen and LNs of ATX mice, we sorted a total of 22'756 memory B cells specific for MARV GPΔMuc (See gating strategy) |
| Gating strategy | Gating strategies for sorting of memory B cells specific for MARV GPΔMuc are in the Figure 2b. |

☒ Tick this box to confirm that a figure exemplifying the gating strategy is provided in the Supplementary Information.

