## [Peer Review File · Nature]

Potent neutralization of Marburg virus by a vaccine-elicited antibody

Corresponding Author: Professor David Veessler

Version 0:

Reviewer comments:

Referee #1

(Remarks to the Author)

The manuscript entitled “Potent neutralization of Marburg virus by a vaccine-elicited monoclonal antibody” by Addetia et al. reports the development of a fully human monoclonal antibody, MARV16, that can neutralize all Marburg variants as well as Ravn virus. Importantly, MARV16 and previously identified RBS-directed antibodies can bind Marburg GP simultaneously which could potentially improve therapeutic treatment options. The authors structural work resolves a portion of the MARV GP1 glycan cap showing that it partially shields the RBS, as is the case for Ebola GPs. Overall, this an excellent study, the work is technically sound, and the paper is very well written.

The main concern with the work is that there is no in vivo data to show that MARV16 can confer protection in an animal model of Marburg infection. Other anti-Marburg human antibodies such as MR191 have been shown to protect rodents and nonhuman primates (NHP) against lethal Marburg disease. In fact, MR191 and MR186-YTE each provided substantial protection when given to NHPs beginning as late as day 5 after virus challenge at an advanced stage of disease. When MR186-YTE was given to NHPs beginning at day 6 after Marburg virus challenge no animals survived as was the case when remdesivir was given as a sole treatment beginning at day 6. However, combining MR186-YTE with remdesivir resulted in 80% protection when the combination treatment was administered to NHPs beginning at day 6. While MR186-YTE was given as a single i.v. treatment remdesivir requires 10-12 daily i.v. injections which is challenging in resource poor settings where Marburg outbreaks tend to occur. Therefore, the authors are correct that there is a clear need to for other antivirals that could be combined with antibodies like MR191 or MR186-YTE to extend the therapeutic window and improve outcome. MARV16 very well may be such an antiviral and would address concerns about escape mutants with single treatments like MR191 or MR186-YTE. It is understood that NHP studies are expensive and hard to do but the authors should do some work in rodent models of Marburg to demonstrate the efficacy of MARV16 and how it compares to the RBS-directed antibodies. Ideally, to show the utility of combining MARV16 with a RBS-directed antibody but at least to show that MARV16 can confer protection.

Referee #2

(Remarks to the Author)

In this manuscript, Addetia et al. describe the stabilization of a class I fusogen protein, GP which is the main fusogen protein for Marburg virus (MARV). The authors characterized GP and used it as an immunogen in humanized transgenic mouse studies which led to the discovery of a neutralizing antibody, MARV16. This antibody was shown to be more potently neutralizing than previously characterized antibodies and was neutralizing against multiple Marburg virus strains. The study then continues to examine the mechanism of antibody-antigen binding through structural studies which allowed for contact mapping and suggestions of the molecular basis for neutralization. Interestingly, MARV16 seems to target a conserved interface on MARV GP which is able to lock the two subunits in a structural conformation resembling the prefusion state. Additionally, the authors provide biophysical and structural evidence that MARV16 can bind concurrently with previously-described RBS-directed antibodies, which could enable the development of a therapeutic antibody cocktail, for example as has previously been done for EBOV (i.e. ZMapp).

Overall, the study follows a logical workflow and is well written. It solidifies multiple established scientific concepts, including the relevance of computationally based immunogen design strategies, the exploration of potent antigenic surfaces in class I fusogens with a focus on prefusion specific states, insights into the position and relevance of fusogen glycan shields, and the discovery of potent neutralizing antibodies from transgenic mice with human Ig repertoire. However, there are several limitations that diminish the overall impact and reach of this work in its current state:

- The structural characterization reveals that MARV16 shares a similar binding mode and mechanism of action to the previously described EBOV GP-directed neutralizing antibody ADI-15946. The cryo-EM map resolved part of the GP1 glycan cap, uncovering an arrangement reminiscent of that observed for EBOV GP. As such, mechanistically, novel insights are limited. In addition, unfortunately, it was found that MARV16 does not broadly cross-react with GPs from the Filovirus genus. An antibody with such broadly neutralizing characteristics would be transformative for the field.
- Two stabilizing mutations (T582P and F583V) were identified using the established ProteinMPNN pipeline that marginally improved the expression yield (~2.5x) and thermostability (~2.6 °C) of a MARV GP Δ Muc construct. It is proposed that these may improve the immunogenicity of MARV vaccines. However, no neutralization is reported for the induced sera from Alloy ATX transgenic mice with human Ig locus. In fact, from 10 binding antibodies recovered in this immunization study, only one showed neutralization potential, bringing into question whether this stabilized immunogen is in fact efficient at eliciting neutralizing responses, or superior in this regard compared to the same construct without the two stabilizing mutations. The authors have in-hand all the reagents and assays established to test this hypothesis to support their claim.
- I realize that the main focus is on MARV16 because of its potent neutralization, but additional information on the complete set of 10 antibodies could provide interesting insights on the molecular mechanisms by which MARV16 is able to stand out as a potent antibody. Since all of the proposed antibodies bound to GP with comparable EC50 values, additional nsEM experiments of the MARV GP Δ Muc ectodomain complexed with various non-neutralizing antibodies would allow for a rough mapping of other binding epitopes that are non-neutralizing. These insights could provide important information for future vaccine design or antibody discovery studies.
- The authors present the molecular rationale for the use of a cocktail of mAbs, but do not demonstrate to what extent these limit the emergence of neutralization escape mutants experimentally, making their argument only conceptual in nature. The authors should perform escape studies (e.g. as previously described in King et al, Cell Host & Microbe 2018) to further support their claim that a cocktail including MARV16 would be superior in limiting viral escape compared to individual antibodies alone.
- Although MARV16 neutralizes more potently than previously-described mAbs, it is unclear whether this leads to better in vivo protection. Marburgvirus challenge of guinea pigs (e.g. as described in Mire et al., Sci. Transl. Med 2017) comparing MARV16 to previously-described antibodies of lower neutralization potency would bolster the claims around superiority for MARV16 and the importance of its discovery. In addition, characterization of whether this antibody is associated with any benefit from effector functions (e.g. human natural killer or dendritic cells or macrophage- or neutrophil-mediated phagocytosis) would also help to better appreciate its protection potential as compared to other antibodies.

Minor comments:

1. The identification and selection of the prefusion stabilizing mutations using ProteinMPNN is poorly described. Please provide the rationale of the computational pipelines, input structures, and potential output values on which the decisions to incorporate these mutations were made.
2. Please provide a residue contact table for the MARV16:GP interaction based on the cryoEM structure. This could help streamline some of the description of contacts in the Results section, which is currently quite extensive.
3. Please consider adding a surface representation of the sequence conservation for GP between the different viral strains or viruses to better understand the breadth of this antibody and potential for this site as a target.
4. The authors state: "All monoclonal antibodies identified to date that target MARV GP display no or weak neutralization potency against MARV GP pseudoviruses and even weaker, if any, activity against authentic MARV (35–38)." Yet, previous publications do report on the neutralization capacity (and mechanistic insights) of antibodies against MARV GP e.g. Hashiguchi et al, Cell 2015; King et al, Cell Host & Microbe 2018; etc. In this statement, to avoid misleading the less familiar readers, it would be important to identify a threshold of potency associated with "weak neutralization" and "potent neutralization" e.g. no antibodies have yet been described that neutralize at <1000 ng/ml in the VSV pseudotyped assay.
5. The supplemental information for the cryoEM data is not sufficient. Please provide data that shows particle distribution and orientation diagnostics which will allow for an in-depth quality analysis of the refined map.
6. The final cryoEM map was refined with an applied C3 symmetry but the model contains differences within the three GP protomers. For example, the glycans on N219 are modeled differently between the three chains. Arguably, the densities for some of the second and third glycan residues are weak and should be considered for removal. At minimum, please review the modeled glycans and confirm their densities for good model to map accuracy.
7. Figure S6D – Please indicate what mask was used for the GSFSC curve and consider adding additional curves for the unmasked and tight masks.
8. "Next, ab-initio 3D reconstruction and heterologous refinement [...]" should be changed to "Next, ab-initio 3D reconstruction and heterogeneous refinement [...]"
9. List the cryoSPARC version for processing in the materials and methods section.
10. Please mention the enforced symmetry for the cryoEM map refinements in the materials and methods section.
11. The PDB model file has inconsistencies in chain and residue assignments. For example, NAG302 in chain C and NAG1 in chain Q should be part of the same chain. This is also causing unwanted chain breaks. Please review the structure model and residue/chain assignments.
12. Figure 1A/3E – Please add start and end residue numbers to the bar representation of the GP constructs.
13. In Fig 1E, given the error bars, are the authors confident in the yield increase to three significant figures?

Version 1:

Reviewer comments:

Referee #1

(Remarks to the Author)

My comments have been appropriately addressed.

Referee #2

(Remarks to the Author)

The authors are to be commended for the additional experiments performed during revisions to substantiate the claims made. All major concerns raised during initial review have been adequately addressed.

Minor point:

-The authors might want to review the immunogen construct nomenclature used for the experiments described in Fig. 1 and beyond. From Fig 1a, it might be interpreted that the MARV GP Δ muc contains the T582P and F583V mutations. Yet, that construct is later called MARV GP Δ mucPV in the same figure. In the next section, it is said that the ATX mice were immunized with MARV GP Δ Muc... I assume this immunogen construct includes the PV mutations? This is not clear based on the nomenclature used for the two constructs (WT and PV stabilized) in Fig. 1h-j.

We would like to thank the reviewers for their time and their valuable feedback.

Ref. 1 (hemorrhagic fever incl. Marburg, vaccines)

The manuscript entitled “Potent neutralization of Marburg virus by a vaccine-elicited monoclonal antibody” by Addetia et al. reports the development of a fully human monoclonal antibody, MARV16, that can neutralize all Marburg variants as well as Ravn virus. Importantly, MARV16 and previously identified RBS-directed antibodies can bind Marburg GP simultaneously which could potentially improve therapeutic treatment options. The authors structural work resolves a portion of the MARV GP1 glycan cap showing that it partially shields the RBS, as is the case for Ebola GPs. Overall, this an excellent study, the work is technically sound, and the paper is very well written.

The main concern with the work is that there is no in vivo data to show that MARV16 can confer protection in an animal model of Marburg infection. Other anti-Marburg human antibodies such as MR191 have been shown to protect rodents and nonhuman primates (NHP) against lethal Marburg disease. In fact, MR191 and MR186-YTE each provided substantial protection when given to NHPs beginning as late as day 5 after virus challenge at an advanced stage of disease. When MR186-YTE was given to NHPs beginning at day 6 after Marburg virus challenge no animals survived as was the case when remdesivir was given as a sole treatment beginning at day 6. However, combining MR186-YTE with remdesivir resulted in 80% protection when the combination treatment was administered to NHPs beginning at day 6. While MR186-YTE was given as a single i.v. treatment remdesivir requires 10-12 daily i.v. injections which is challenging in resource poor settings where Marburg outbreaks tend to occur. Therefore, the authors are correct that there is a clear need to for other antivirals that could be combined with antibodies like MR191 or MR186-YTE to extend the therapeutic window and improve outcome. MARV16 very well may be such an antiviral and would address concerns about escape mutants with single treatments like MR191 or MR186-YTE. It is understood that NHP studies are expensive and hard to do but the authors should do some work in rodent models of Marburg to demonstrate the efficacy of MARV16 and how it compares to the RBS-directed antibodies. Ideally, to show the utility of combining MARV16 with a RBS-directed antibody but at least to show that MARV16 can confer protection.

We agree with the Reviewer on the importance of demonstrating that MARV16 can confer protection *in vivo*, and we have therefore conducted a therapeutic challenge study in guinea pigs using guinea pig-adapted MARV. Animals were treated with MARV16 1, 2 or 4 days following infection and monitored for 29 days. We observed significant improvements in survival for the MARV16-treated animals (50%, 83%, and 60% survival for animals receiving MARV16 1, 2, or 4 dpi, respectively) compared to animals administered an isotype control (0% survival). These data indicate that MARV16 provides therapeutic protection against stringent MARV challenge (at a dose per kg 5x lower than that used for MR186-YTE evaluation in NHPs, PMID: 33767178) and are presented in Fig 5 and the “*MARV16 protects against MARV section*” of the revised manuscript.

We agree that comparing the protective efficacy of MARV16 to RBS-directed antibodies as well as showing the protective efficacy of a cocktail of MARV16 and an RBS-directed antibody are worthy of future study but are beyond the scope of this paper. However, as detailed below, we are showing in the revised manuscript that antibody cocktails combining MARV16 and an RBS-directed antibody increase the barrier for emergence of escape mutants.

Figure 5. MARV16 protects guinea pigs from stringent MARV challenge. **a)** Schematic of the MARV challenge study assessing the therapeutic efficacy of MARV16. IP: intraperitoneal. **b)** Survival curves for guinea pigs (n = 6 per group) challenged with 1,000 PFU of MARV/Angola and treated with MARV16 or an isotype control antibody. Animals were monitored for 29 dpi. Survival curves for the groups administered MARV16 1 dpi and 4 dpi excluding the animals with undetectable plasma MARV16 concentration one day after treatment are shown as boxes with dashed lines. Statistical differences in survival between groups were only assessed including all animals per group and using the Kaplan-Meier survival analysis and Holm-Šídák correction comparing isotype control antibody-treated guinea pigs to the MARV16-treated groups. **c)** MARV/Angola viral loads measured in the plasma of infected guinea pigs at 5 dpi. The black line indicates the geometric mean viral load for each group and the dotted line denotes the limit of detection (viral load $\leq 2 \times 10^3$ GE/mL). Statistical differences in viral loads between groups were assessed using the Kruskal-Wallis test with Dunn's post-test comparing the isotype control antibody-treated guinea pigs to the MARV16-treated groups. **d-f)** Daily body weights (d), body temperatures (e), and clinical scores (f) for the guinea pigs for the duration of the challenge study. Animals with undetectable plasma concentrations of MARV16 one day after treatment are denoted with triangles indicating that in these animals MARV16 was sequestered at the injection site and cleared or 'soaked up' by the challenge virus immediately before reaching the blood stream.

Ref. 2 (structural biol., vaccines, Ab-Ag)

In this manuscript, Addetia et al. describe the stabilization of a class I fusogen protein, GP which is the main fusogen protein for Marburg virus (MARV). The authors characterized GP and used it as an immunogen in humanized transgenic mouse studies which led to the discovery of a neutralizing antibody, MARV16. This antibody was shown to be more potently neutralizing than previously characterized antibodies and was neutralizing against multiple Marburg virus strains. The study then continues to examine the mechanism of antibody-antigen binding through structural studies which allowed for contact mapping and suggestions of the molecular basis for neutralization. Interestingly, MARV16 seems to target a conserved interface on MARV GP which is able to lock the two subunits in a structural conformation resembling the prefusion state. Additionally, the authors provide biophysical and structural evidence that MARV16 can bind concurrently with previously-described RBS-directed antibodies, which could enable the development of a therapeutic antibody cocktail, for example as has previously been done for EBOV (i.e. ZMapp).

Overall, the study follows a logical workflow and is well written. It solidifies multiple established scientific concepts, including the relevance of computationally based immunogen design strategies, the exploration of potent antigenic surfaces in class I fusogens with a focus on prefusion specific states, insights into the position and relevance of fusogen glycan shields, and the discovery of potent neutralizing antibodies from transgenic mice with human Ig repertoire. However, there are several limitations that diminish the overall impact and reach of this work in its current state:

- The structural characterization reveals that MARV16 shares a similar binding mode and mechanism of action to the previously described EBOV GP-directed neutralizing antibody ADI-15946. The cryo-EM map resolved part of the GP1 glycan cap, uncovering an arrangement

reminiscent of that observed for EBOV GP. As such, mechanistically, novel insights are limited. In addition, unfortunately, it was found that MARV16 does not broadly cross-react with GPs from the Filovirus genus. An antibody with such broadly neutralizing characteristics would be transformative for the field.

We agree that an antibody capable of neutralizing all filoviruses potentially would be truly remarkable, however, we speculate that identifying such antibody may be very difficult given that sera collected from individuals who received Ebola viruses vaccines (PMID: 40173257) or sera or B cells from individuals infected with either SUDV, BDBV or MARV (PMID: 27335383, 30728263) do not display broad cross-reactivity across different filoviruses. MARV16 broadly neutralizes VSV pseudotyped with filoviruses as divergent as Dehong virus GP, which shares only 49.9% amino acid identity with the MARV/Musoke GP (Figure 3g-h), underscoring the exceptional breadth of this antibody given this large genetic divergence. The comparison to ADI-15946 does not reduce the novelty of our findings given that EBOV and MARV are entirely distinct pathogens, as demonstrated by the fact that none of the vaccines or antibodies developed for EBOV are relevant for MARV. Our results establish MARV16 as a best-in-class MARV neutralizing antibody in terms of potency and breadth (Figure 3), which is already in consideration for future clinical advancement by multiple interested parties.

- Two stabilizing mutations (T582P and F583V) were identified using the established ProteinMPNN pipeline that marginally improved the expression yield (~2.5x) and thermostability (~2.6 °C) of a MARV GP Δ Muc construct. It is proposed that these may improve the immunogenicity of MARV vaccines. However, no neutralization is reported for the induced sera from Alloy ATX transgenic mice with human Ig locus. In fact, from 10 binding antibodies recovered in this immunization study, only one showed neutralization potential, bringing into question whether this stabilized immunogen is in fact efficient at eliciting neutralizing responses, or superior in this regard compared to the same construct without the two stabilizing mutations. The authors have in-hand all the reagents and assays established to test this hypothesis to support their claim.

As requested, we conducted an immunogenicity study in which mice were immunized with 3 doses of 0.1, 1, or 10 μ g of MARV GP Δ Muc or MARV GP Δ Muc_{PV} spaced four weeks apart and sera was collected 2 weeks after each immunization (cf revised Figure 1h-j). We observed increased GP binding titers for the MARV GP Δ Muc_{PV}-immunized mice, however, the neutralizing antibody titers for both groups were similarly modest. These data indicate the MARV GP Δ Muc_{PV} ectodomain is more immunogenic than the MARV GP Δ Muc, which might lead to superior protection given the importance of Fc-mediated effector functions for protection against MARV mediated by current vaccines (given that they hardly elicit neutralizing antibodies) (PMID: 36516269; 30567978; 25225676). However, this suggests that alternative strategies for delivering the stabilized GP will likely be necessary to elicit robust neutralizing antibody titers (e.g. nanoparticle display, mRNA,...). Given that the main focus of this manuscript is the evaluation of a best-in-class therapeutic mAb against MARV, future development of MARV vaccine candidates will be the focus of future studies.

- I realize that the main focus is on MARV16 because of its potent neutralization, but additional information on the complete set of 10 antibodies could provide interesting insights on the molecular mechanisms by which MARV16 is able to stand out as a potent antibody. Since all of the proposed antibodies bound to GP with comparable EC50 values, additional nsEM experiments of the MARV GP Δ Muc ectodomain complexed with various non-neutralizing antibodies would allow for a rough mapping of other binding epitopes that are non-neutralizing. These insights could provide important information for future vaccine design or antibody discovery studies.

Per the Reviewer's request, we examined the epitopes recognized by the 9 non-neutralizing antibodies discovered by first performing an epitope binning experiment to identify binding groups (cf revised Extended Data Figure 7). Binding group 1 consists of MARV4, MARV12, MARV18, MARV21, and MARV23, binding group 2 consists of MARV11 and MARV14, and MARV7 and MARV20 each belong to unique binding groups. We then used negative stain electron microscopy to identify the epitopes for each binding group. Binding group 1, represented by MARV18, binds to HR2, while both binding group 2 antibodies, represented by MARV14, and MARV7 bind the GP2 wing. We were unable to resolve the epitope recognized by MARV20 as the 2D class averages suggest that it binds to a flexible region of the GP.

Extended Data Figure 7. Identification of MARV GP Δ Muc antigenic sites recognized by non-neutralizing monoclonal antibodies. (a-b) Percent binding (a) to the MARV GP Δ Muc and BLI

traces (b) for antibody pairs evaluated in the epitope binning experiments. Antibody pairs showing reciprocal blocking relationships were classified as belonging to the same binding group. Data from one biological replicate is shown and representative of two biological replicates. c-e, 3D reconstructions, representative 2D class averages, and angular distribution plots obtained by single particle electron microscopy analysis of negatively stained MARV GP Δ Muc bound to the MARV18 (c), MARV14 (d), or MARV7 (e) Fabs. (f) Representative 2D classes from an electron microscopy dataset of negatively-stained MARV GP Δ Muc-MARV20 Fab complex.

- The authors present the molecular rationale for the use of a cocktail of mAbs, but do not demonstrate to what extent these limit the emergence of neutralization escape mutants experimentally, making their argument only conceptual in nature. The authors should perform escape studies (e.g. as previously described in King et al, Cell Host & Microbe 2018) to further support their claim that a cocktail including MARV16 would be superior in limiting viral escape compared to individual antibodies alone.

As requested, we performed antibody escape experiments (Figure 6d-g) and found that antibody cocktails comprised of MARV16 and MR78 or MARV16 and MR191 required two or more mutations for escape and that these mutations map to the distinct antigenic sites recognized by each antibody in the cocktail. These data support the claim that a cocktail composed of MARV16 and an RBS-directed antibody is superior to limiting viral escape than individual antibodies alone. These data are presented in the “*Formulation of MARV neutralizing antibody cocktails*” and Fig6 of the revised manuscript.

Figure 6. Formulation of a MARV monoclonal antibody cocktail. **a)** Competitive binding assay of MARV16, MR78, and MR191 IgG to the MARV16-bound MARV GP Δ Muc ectodomain using BLI. Data presented are from one biological replicate and are representative of data from two biological replicates using distinct batches of protein. **b-c)** Representative 2D classes and 3D reconstruction of negatively stained MARV GP Δ Muc ectodomain bound to MR78 and MARV16 Fabs (b) or MR191 and MARV16 Fabs (c). The position of the MR78 (pink) or MR191 (blue) Fabs were determined by superimposing the RAVV GP-MR78 Fab (PDB: 5UQY) or RAVV GP-MR191 Fab (PDB: 6BP2) structures with our MARV GP-MARV16 Fab structure. **d-f)** Escape mutations identified for MARV16 alone (d), the MARV16-MR78 antibody cocktail (e), or the MARV16-MR191 antibody cocktail (f) using replication-competent VSV encoding the MARV/Musoke GP instead of VSV G. Two selection experiments were performed using the separately plaque-purified VSV-MARV/Musoke GP isolates 2B2 and 2B4. The virus was passaged in increasing concentrations of antibody until the virus showed significant cytopathic effect (>20%) in the presence of 100 μ g/mL of antibody. Mutations were identified by deep sequencing of the viral supernatant

and those that reached a frequency of at least 10% percent are displayed in the plots and colored by the isolate that they were identified from. The V547G mutation that was also identified in 2B2 passaged without antibody is displayed in gray. **g**) Escape mutations (red) identified during the antibody selection experiments shown on the surface of the MARV GP Δ Muc (gray). The position of the MR78 (pink) was determined by superimposing the RAVV GP-MR78 Fab (PDB: 5UQY) structure with our structure of MARV GP in complex with the MARV16 Fab (green). C226, F447S, and L448P were not resolved in our structure and not displayed on the MARV GP.

- Although MARV16 neutralizes more potently than previously-described mAbs, it is unclear whether this leads to better in vivo protection. Marburgvirus challenge of guinea pigs (e.g. as described in Mire et al., Sci. Transl. Med 2017) comparing MARV16 to previously-described antibodies of lower neutralization potency would bolster the claims around superiority for MARV16 and the importance of its discovery. In addition, characterization of whether this antibody is associated with any benefit from effector functions (e.g. human natural killer or dendritic cells or macrophage- or neutrophil-mediated phagocytosis) would also help to better appreciate its protection potential as compared to other antibodies.

Thank you for this suggestion. Please see above answer to Reviewer 1 for results from the challenge study.

We evaluated the ability of MARV16 to trigger activation of Fc γ RIIa and Fc γ RIIIa, surrogate assays for ADCP and ADCC, respectively (Extended Data Figure 8). MARV16 did not activate either Fc γ RIIa or Fc γ RIIIa, suggesting the observed protection in the challenge study originates solely from the antibody's direct neutralizing activity.

Minor comments:

1. The identification and selection of the prefusion stabilizing mutations using ProteinMPNN is poorly described. Please provide the rationale of the computational pipelines, input structures, and potential output values on which the decisions to incorporate these mutations were made.

We focused on identifying stabilizing mutations in GP2 HR1_C as this region is metastable in the prefusion conformation and undergoes rearrangement from a loop in the prefusion conformation to a helix in the postfusion conformation. Additionally, previous studies on the EBOV GP have identified mutations in HR1_C that stabilize the prefusion conformation suggesting HR1_C mutations would stabilize the prefusion MARV GP conformation as well (PMID: 32234486; 33976149). We have previously used ProteinMPNN to stabilize other viral glycoproteins including the Langya virus G and Epstein-Barr virus gB proteins (PMID: 38593070; 39484573), motivating the use of ProteinMPNN for identifying stabilizing mutations for MARV GP. The previously determined RAVV GP structure (PDB: 6BP2; PMID: 29324225) was used as an initial model for ProteinMPNN and the identified mutations were visually examined for their compatibility in the prefusion trimer prior to incorporation into the MARV GP Δ Muc ectodomain. These details have been added to the "*Identification of stabilizing mutations in MARV GP2 HR1_C*" section.

2. Please provide a residue contact table for the MARV16:GP interaction based on the cryoEM structure. This could help streamline some of the description of contacts in the Results section, which is currently quite extensive.

We have added a residue contact table as Extended Data Table 3 and simplified the text accordingly, as requested.

3. Please consider adding a surface representation of the sequence conservation for GP between the different viral strains or viruses to better understand the breadth of this antibody and potential for this site as a target.

As requested, we have added a surface representation showing the sequence conservation for the GP residues comprising the MARV16 epitope across the *Marburgvirus* isolates (MARV variants and RAVV) assessed in this study (Extended Data Figure 6a).

4. The authors state: “All monoclonal antibodies identified to date that target MARV GP display no or weak neutralization potency against MARV GP pseudoviruses and even weaker, if any, activity against authentic MARV (35–38).” Yet, previous publications do report on the neutralization capacity (and mechanistic insights) of antibodies against MARV GP e.g. Hashiguchi et al, Cell 2015; King et al, Cell Host & Microbe 2018; etc. In this statement, to avoid misleading the less familiar readers, it would be important to identify a threshold of potency associated with “weak neutralization” and “potent neutralization” e.g. no antibodies have yet been described that neutralize at <1000 ng/ml in the VSV pseudotyped assay.

We have updated the text with thresholds for potency as follows:

“All monoclonal antibodies identified to date that target MARV GP display no or weak neutralization potency (>1,000 ng/mL) against MARV GP pseudoviruses and even weaker, if any, activity (>100 µg/mL) against authentic MARV (36–39)”

5. The supplemental information for the cryoEM data is not sufficient. Please provide data that shows particle distribution and orientation diagnostics which will allow for an in-depth quality analysis of the refined map.

We have added the viewing direction distribution and conical FSC plots to Extended Data Figure 5, as requested.

6. The final cryoEM map was refined with an applied C3 symmetry but the model contains differences within the three GP protomers. For example, the glycans on N219 are modeled differently between the three chains. Arguably, the densities for some of the second and third glycan residues are weak and should be considered for removal. At minimum, please review the modeled glycans and confirm their densities for good model to map accuracy.

We have updated the model to reflect the applied C3 symmetry for the cryoEM map. We reviewed the density for the modeled glycan and respectfully disagree with the Reviewer. The modeled glycans have strong support in the unsharpened map (see below).

7. Figure S6D – Please indicate what mask was used for the GSFSC curve and consider adding additional curves for the unmasked and tight masks.

The mask_fsc was used for the GSFSC curve. This information and the unmasked and tight masks have been added to the figure.

8. "Next, ab-initio 3D reconstruction and heterologous refinement [...]" should be changed to "Next, ab-initio 3D reconstruction and heterogeneous refinement [...]".

Fixed.

9. List the cryoSPARC version for processing in the materials and methods section.

The cryoSPARC version has been added.

10. Please mention the enforced symmetry for the cryoEM map refinements in the materials and methods section.

We apologize for this omission. The enforced C3 symmetry has been added to the methods section.

11. The PDB model file has inconsistencies in chain and residue assignments. For example, NAG302 in chain C and NAG1 in chain Q should be part of the same chain. This is also causing unwanted chain breaks. Please review the structure model and residue/chain assignments.

We have updated the model and fixed inconsistencies in the chain and residue assignments.

12. Figure 1A/3E – Please add start and end residue numbers to the bar representation of the GP constructs.

Residue numbers have been added to Fig3e, as requested.

13. In Fig 1E, given the error bars, are the authors confident in the yield increase to three significant figures?

We have modified the values in Figure 1e to display two significant figures instead of three. Additionally, we have included the fold-changes in yield relative to the MARV GPΔMuc ectodomain for each of the four biological replicates in the table below to support presentation of two significant figures.

Fold-change in yield of stabilized MARV GPΔMuc ectodomains compared to the base MARV GPΔMuc ectodomain.

MARVΔMuc	T582P	F583V	TF582PV
1	1.42	1.34	1.84
1	1.40	1.88	2.88
1	1.85	1.87	2.33
1	1.83	1.51	3.09